# Photothermal/Photoacoustic Therapy Combined with Metal-Based Nanomaterials for the Treatment of Microbial Infections

**DOI:** 10.3390/microorganisms11082084

**Published:** 2023-08-14

**Authors:** Nour Mammari, Raphaël E. Duval

**Affiliations:** 1Université de Lorraine, CNRS, L2CM, F-54000 Nancy, France; 2ABC Platform®, F-54505 Vandœuvre-lès-Nancy, France

**Keywords:** photothermal therapy, photoacoustic, antimicrobial strategy, metal-based nanoparticles

## Abstract

The increased spread and persistence of bacterial drug-resistant phenotypes remains a public health concern and has contributed significantly to the challenge of combating antibiotic resistance. Nanotechnology is considered an encouraging strategy in the fight against antibiotic-resistant bacterial infections; this new strategy should improve therapeutic efficacy and minimize side effects. Evidence has shown that various nanomaterials with antibacterial performance, such as metal-based nanoparticles (i.e., silver, gold, copper, and zinc oxide) have intrinsic antibacterial properties. These antibacterial agents, such as those made of metal oxides, carbon nanomaterials, and polymers, have been used not only to improve antibacterial efficacy but also to reduce bacterial drug resistance due to their interaction with bacteria and their photophysical properties. These nanostructures have been used as effective agents for photothermal therapy (PTT) and photodynamic therapy (PDT) to kill bacteria locally by heating or the controlled production of reactive oxygen species. Additionally, PTT or PDT therapies have also been combined with photoacoustic (PA) imaging to simultaneously improve treatment efficacy, safety, and accuracy. In this present review, we present, on the one hand, a summary of research highlighting the use of PTT-sensitive metallic nanomaterials for the treatment of bacterial and fungal infections, and, on the other hand, an overview of studies showing the PA-mediated theranostic functionality of metal-based nanomaterials.

## 1. Introduction

The widespread occurrence of infections caused by multidrug-resistant (MDR) bacteria is increasing around the world [1,2,3]. Since the first reports of drug resistance on Enterobacteriaceae in the 1950s, numerous bacterial strains with drug-resistant phenotypes have been reported [3,4]. The increase in the frequency, magnitude, spread, and persistence of drug-resistant bacterial phenotypes has contributed significantly to the challenge. This is due to the misuse of antibacterial agents and the emergence of resistance to antibiotics not only in nosocomial settings but also in community ones [5]. European Union/European Economic Area (EU/EEA) statistical data show that each year more than 670,000 infections are due to antibiotic-resistant bacteria, which cause 33,000 deaths [5]. The emergence of new resistance mechanisms in these bacterial strains contributes to the ineffectiveness of existing antibiotics, to the prolongation of the disease, and therefore of the hospitalization, resulting in an increase in expenditure. This alarming situation requires new strategies to fight against antimicrobial resistance (AMR).

Nanotechnology-based drug delivery systems are considered promising strategies in combating bacterial infections, and are expected to improve therapeutic efficacy and minimize side effects [6]. Evidence and patents show that various nanomaterials with intrinsic antibacterial properties are being developed: metal-based (e.g., silver, gold, copper, and zinc oxide) nanoparticles (NPs) with intrinsic antibacterial properties, have also been widely used to not only improve antibacterial efficiency but also reduce bacterial drug resistance due to their high surface contact and interaction with bacteria [7,8,9,10]. In addition, the tunable shape and functional modification of nanomaterials also endow them with ideal capability for the delivery of antibacterial drugs or bioactive molecules to wound or infection sites, thus promoting treatment efficacy [11].

Unfortunately, conventional nanodrug delivery systems have been shown to present practical dilemmas, including the method of nanoparticle delivery (which could be incomplete or slow), incomplete effectiveness in targeting infected sites, and low-capacity biofilm penetration [12]. In addition, the incorporation of metal ions like Ag^+^, Cu^2+^, Zn^2+^, Co^2+^, Fe^3+^, and Pb^2+^ could lead to an increase in the innate cytotoxicity response of these materials and cause a synergistic effect with the bacterial infection [13]. Stimuli-responsive nanomaterials are hence developed to overcome the disadvantages of conventional nanoparticles [14].

Furthermore, the particular photophysical properties of some of the nanomaterials, such as those made of metal oxides [15], carbon-based nanomaterials [16], and polymers [17], have allowed them to be effective agents for photothermal therapy (PTT) and photodynamic therapy (PDT), killing bacteria locally by heating or controlled production of reactive oxygen species (ROS) in the infected sites without the problem of bacterial resistance [18].

Phototherapy (PT) has its origin in the work of Niels Finsen, who was awarded the Nobel Prize for successfully treating smallpox and cutaneous tuberculosis using red and UV light in 1903 [19]. Among the types of phototherapies, photodynamic therapy is the most commonly studied technique. PDT or photodynamic inactivation (PDI) is a means of phototherapy combining light, a substrate, and a photosensitizer to generate free radicals and ROS to eradicate undesired cells [20,21]. In recent years, photothermal agents (PTAs) with high light-to-heat conversion ability have been developed due to their broad prospects for clinical application due to the fact that PTT is capable of damaging the integrity of pathogenic bacteria [22]. Hyperthermia could destroy the structure of a biofilm by inactivating its inherent bioactive matrices and thus facilitate the penetration of antibacterial or antifungal agents [23]; this has been determined in in vitro and in vivo studies against *Candida albicans* [24] and Gram-negative drug-resistant/multidrug-resistant isolates of *Acinetobacter baumannii*, *Klebsiella pneumoniae*, and *Pseudomonas aeruginosa*, and against clinical strains of methicillin-resistant *Staphylococcus aureus* (MRSA) [25]. It has been reported that metal nanoparticles coupled with polymers are an effective approach to removing bacterial biofilm in order to internalize and disrupt surface functionalization, to increase biocompatibility. By coupling this treatment to PDT, the latter could act by killing the bacteria hidden in the biofilm via the destruction of the channels engaged in the transport of nutrients towards the central region and the loss of the integrity of the biofilm. However, these photothermal effects presented some side effects by damaging normal tissues. The combination of PDT and PTT has emerged to maximize efficacy while minimizing side effects [26]. Moreover, PTT destroys the structure and composition of biofilms through the physical damage of localized hyperthermia, with minimal systemic toxicity and without inducing bacterial resistance [27]. To achieve desirable curative efficiency and minimal side effects, synergistic therapeutic strategies combining PTT with other techniques, such as chemotherapy, photocatalytic therapy (PCT), immunotherapy, sonodynamic therapy (SDT), etc. [28], have recently been brought into focus. Additionally, multifunctional nanoplatforms with targeted diagnostic and therapeutic imaging functionality (i.e., theranostics) are of great interest in the field of precision nanomedicine. PTT or PDT has also been combined with photoacoustic (PA) imaging [29]. In photoacoustic imaging, plasmonic nanoparticles are widely used. They lead to a strong light/matter interaction. It was determined that the absorption efficiency of metallic nanoparticles should be higher than that of dye molecules. This results in the efficient generation of photoacoustic signals. These signals are generated by exciting either the signals of the constituent molecules of living cells or those of exogenous contrast agents. However, the optimal excitation light wavelength can be selected depending on the size, shape, and composition of the nanoparticle in question. Moreover, this therapy could be targeted by the control of the surface chemistry which confers a selective functionalization of the metallic nanoparticle [30].

Many existing materials have been exploited as photothermal agents, including transition metal oxide/sulfide nanomaterials (e.g., manganese dioxide (MnO_2_) and copper monosulfide (CuS)) [31], metal nanostructures (e.g., CuNPs, gold (Au) NPs, and silver (Ag) NPs) [32,33,34], and carbon-based materials (e.g., carbon nanotubes (CNTs) and graphene) [35,36].

Taking the example of Cu_2_O NPs, the study by Yang et al. (2021) [29] showed them to be a theranostic agent helping to distinguish infected cells from normal cells [29]. Under the effect of near-infrared (NIR) irradiation and photoacoustic imaging, Cu_2_O NPs can effectively catalyze H_2_O_2_ at the site of infection to produce hydroxyl radicals (•OH) with strong antibacterial properties via a Fenton-like reaction, resulting in bacterial cell membrane damage [29]. The principle of photoacoustics imaging is based on the conversion of photon energy into acoustic pressure waves to acquire images. It has emerged as a new non-ionizing imaging modality, which allows deep tissue penetration compared to optical imaging due to the reduced scattering of acoustic waves compared to light [37]. Accordingly, the design of an antimicrobial agent for PTT could be guided by PA imaging and simultaneously improve treatment efficacy, safety, and accuracy.

In this review, we would first like to provide a summary of the research highlighting the use of PTT-responsive metallic nanomaterials for the treatment of bacterial and fungal infections, and secondly an overview of studies showing the PA-mediated theranostic functionality of metal-based nanomaterials.

## 2. Methodology

In this study, an in-depth bibliographic search was carried out in order to scan all the studies involving the use of metal-based nanomaterials stimulated by PTT in the treatment of bacterial and fungal infections. The literature search was stopped on 28 July 2023 using the Medline and PubMed databases, and the following keywords: ‘bacteria’, ‘photothermal therapy’, and ‘nanoparticles’. Out of a total of 403 publications, 40 reviews were excluded. Then, 88 articles using non-metallic nanomaterials and 80 treating viral or tumor cells were excluded as well as 1 article in which authors did not use PTT. In addition, 82 articles that did not use gold and silver nanoparticles (Fe:20; Cu:28; Ti:9; Zn:4; Te:2; Pt:3; Rh:2; Se:3; PB:1; Pd:1; Re:1; and BP:3) were excluded. In the end, 113 articles were included in the study (Figure 1).

Moreover, an additional bibliographic study was carried out to show the use of metal-based nanomaterials in PA imaging. Using the same databases and the keywords ‘bacteria’, ‘photoacoustic’, and ‘nanoparticles’, a total of 49 articles were identified. A total of 4 reviews were excluded, as well as 27 articles that did not use metal-based nanomaterials or gold or silver NPs, and in which treatments were performed on tumor cells. In the end, 11 studies were included (Figure 2).

## 3. PTT Combined with Metal or Metal Oxide Nanoparticles as Antibacterial Treatment Agents

The results of the bibliographic search of this part are summarized in Table 1.

### 3.1. Silver Nanoparticles (AgNPs)

Silver nanoparticles (AgNPs) (220 nm) have been widely studied as a treatment for antibiotic-resistant bacterial infections. Unfortunately, the evidence has not yet demonstrated a significant efficiency of a nanoparticle synthesis protocol. However, nanoparticles can be incorporated into inorganic compounds like silica, quantum dots, metal nanoparticles, and metal oxide nanoparticles or into organic complexes such as liposomes, micelles, dendrimers, and polymeric nanoparticles to maximize effectiveness against bacterial cells. The microbial effect of AgNPs is mediated by their incursion abilities into microbial structures and their interaction with the bacterial membrane. Ag^+^ ions can create pores in the bacterial cell wall by binding with the sulfur- and phosphorus-containing proteins of the cell wall and the cell membrane [90,91], and then generate oxidative stress, and ultimately cause cell death [7]. The potential antibacterial activity of black phosphorus (BP) nanosheets decorated with AgNPs was evaluated. The photothermal effect of BP nanosheets enabled Ag@BP nanohybrids (1 mg/kg) to rapidly disrupt the bacterial membrane of MRSA in a mouse model and minimize tissue damage associated with infection, under NIR light irradiation at 808 nm (0.8 W/cm^2^) for 5 min. Indeed, the released Ag+ increases oxidative stress and sustainably suppresses bacterial proliferation for a long period [38]. The researchers were able to demonstrate that making this molecule sensitive to light increases its effectiveness under the effect of photonic and thermal irradiation. However, AgNPs have also been tested alone as light-exciting antibacterial agents. Here, a study demonstrated that triangular AgNPs, excited by a NIR laser at 808 nm (1.3 W/cm^2^) for 10 min, have the ability to effectively kill *E. coli*, extended-spectrum β-Lactamase-producing *E. coli* (ESBLE), *S*. *aureus,* and MRSA both in vitro and in vivo with high biocompatibility and biosafety [39].

The photothermal effect of AgNPs (<10 nm) incorporated into a polymer complex was evaluated in many studies. Quaternization (DQC)-stabilized AgNPs formed nanomicelles of quaternized chitin incorporated with AgNPs (DQCA). The photothermal conversion efficiency of the polymer DQCA was up to 65% at 660 nm, 1.0 W/cm^2^ for 10 min. The DQCA is characterized by having bactericidal and antibiofilm activities combined with superior photothermal capacity.

The in vitro antibacterial rate of DQCA accompanied with NIR laser irradiation (660 nm, 1.0 W/cm^2^) was up to 95% in 10 min. DQCA (0.015 mg/mL) showed antibacterial rates of 99.5% for *E. coli* and 95.7% for *S. aureus*, respectively. The eradication efficiency against both the *E. coli* and *S. aureus* biofilms reached up to 99.9%. Moreover, in vivo experiments of *S. aureus*-biofilm-infected wound healing assay in a mouse model demonstrated that the combined effect of DQCA nanomicelle can significantly accelerate wound healing, essentially when the treatment is combined with NIR irradiation (wound healing was improved by more than 2.5% compared to the name treatment combined with NIR irradiation) simultaneously reducing inflammation, improving re-epithelialization, and promoting collagen deposition, with an excellent cytocompatibility [40].

In the same context, a three-in-one bactericidal flower-shaped nanocomposite was assembled using Ag nanoparticles/nano-flowers of phosphotungstic/polyoxometalate/acid-polydopamine (AgNPs/POM-PDA). The nanocomposite (POM-PDA: 6.4 ± 0.8 μm; AgNPs: 50 nm) exhibited photothermal therapy at a concentration of 200 μg/mL when exposed to an NIR light 808 nm laser at 0.75 W/cm^2^ for 6 min via photothermal conversion by PDA (60.2 °C). The principle of this treatment is to combine the effect of Ag+ ions released by AgNPs by photothermia with chemodynamic therapy (CDT) via a POM catalytic Fenton-type reaction. This combined PTT/CDT/Ag^+^ treatment had a good yield in terms of antibacterial activity against *E. coli* and *S. aureus*. The flower-shaped AgNPs/POM-PDA nanocomposite was then embedded in a biocomposite chitosan (CS)/gelatin (GE) hydrogel to have an effective multifunctional dressing for wound healing and biocompatibility [41].

PDA@Ag nanoparticles (290 nm) are synthesized via the growth of Ag on the surface of PDA nanoparticles and then encapsulated in a cationic guar gum (CG) hydrogel network. The η of PDA@Ag (10 mg/mL)) and CG/PDA@Ag was calculated to be 36.1% and 38.2%. Here, the in vitro antibacterial activity of the CG/PDA@Ag hydrogel was explored against *E. coli* and *S. aureus*. After NIR irradiation, the temperature of the bacterial suspension embedded in CG/PDA@Ag (440 nm) hydrogel 2 (CPA2) increased significantly from 37 to 67.3 °C under an 808 nm NIR laser (1 W/cm^2^, 3 min). The formulated CG/PDA (80:2 called CP2) and CG/PDA@Ag (80:0 and 80:2, called CG and CPA2, respectively) hydrogels were used in an experimental trial. The trial was performed in vivo, and the rats were randomly assigned to six groups: phosphate-buffered saline (PBS), CG, CP2, CPA2, CP2 + NIR, and CPA2 + NIR. The results showed that in these active groups, the CG/PDA@Ag can capture and kill bacteria through effective interactions between the hydrogel and bacteria, thereby exerting an antibacterial effect. In addition, a live/dead cell test was performed via the SYTO9/PI method (green color for living cells, and red color for dead cells). The results showed that the two bacterial strains *S. aureus* and *E. coli* treated with PBS showed noticeable green fluorescence, and only a few bacteria were stained with red fluorescence. By contrast, some red fluorescence appeared in the CG hydrogel treatment group. In addition, the red color was gradually becoming more prominent after the PDA@Ag-doped CG hydrogel treatment, and the CPA2 + NIR treatment group showed the strongest red fluorescence, reflecting the synergistic eradication effect on *S. aureus* and *E. coli*. The in vivo antibacterial potential of the developed CG/PDA@Ag hydrogel was evaluated using a bacteria-infected rat model. After 3 min of NIR irradiation at 808 nm (1.0 W/cm^2^), the temperature of the wound region of the CG/PDA@Ag (CPA2) hydrogel was raised to 56.2 °C. However, the CG/PDA@Ag + NIR group was found to have superior antibacterial efficacy both in vitro and in vivo, and biocompatibility [23].

Another hydrogel model constructed based on PDA nanoparticles is a PDA-inked Ag nanozyme-based bilayer injectable hydrogel [42]. The PDA@AgNPs@bilayer (spherical AgNPs, 80–100 nm) had greater effects on bacterial growth inhibition under the effect of Ag+ released following NIR irradiation at 808 nm for 10 min, 1.5 W/cm^2^. Moreover, a hybrid hydrogel was prepared by incorporating borax into a mixture of 3-aminophenylboronic-acid-grafted sodium alginate and nano-silver-decorated polydopamine nanoparticles (SABA/Borax/PDA@AgNPs) [44]. For PDA@AgNPs@bilayer, hyperthermia enhanced peroxidase activity (POD-like) to produce hydroxyl radicals (•OH), which give the hydrogel excellent antibacterial properties when combined with the released Ag+. Bactericidal activity has been reported for the PDA@AgNPs@bilayer stimulated by NIR irradiation. It was attributed particularly to the combined actions of the intrinsic bactericidal activity of Ag+ and PTT [42]. For the hybrid hydrogel SABA/Borax/PDA@AgNPs, 808 nm NIR light irradiation for 5 min at low-temperature PTT (LT-PTT, ≤45 °C) stimulates the photothermal property of polydopamine nanoparticles. The broad-spectrum antibacterial activity of AgNPs and the photothermal property of polydopamine nanoparticles result in the excellent antibacterial activity of SABA/Borax/PDA@AgNPs in vitro against *S. aureus* and *E. coli* as well as in an in vivo in mouse skin wound model without distinct tissue injury. Cytocompatibility evaluation demonstrated a high cell viability [44].

Polydopamine nanoparticles were also used to construct an antibacterial fibrous membrane consisting of electrospun poly (polycaprolactone) scaffolds and MXene/Ag_3_PO_4_ bio-heterojunctions coated with polydopamine (PDA) (MX@AgP bio-HJ). This polymer was designed to be an antibacterial treatment and to aid wound healing in vivo under PTT stimulation. Upon near-infrared illumination for 10 min at 808 nm (1.5 W/cm^2^), the MX@AgP (NP) nanoparticles (0.04 mg/L) in the nanofiber electrospun membranes exert a bactericidal effect against *S. aureus* (ATCC 25923) and *E. coli* (ATCC 25922) through phototherapy and release Ag+ ions which prevent the remaining bacteria from multiplying. Indeed, the in vivo results showed the efficiency of the photoactivated nanofibrous membrane with an antibacterial power while remodeling an infected wound microenvironment into a regenerative microenvironment [32].

The effectiveness of AgNPs was also determined using a multifunctional hydrogel designed based on polyoxometalate (AgPOM, 70 nm) nanoparticles derived from Ag-doped Mo_2_C, urea, gelatin, and tea polyphenols (TPs) for antibacterial and healing acceleration. After being injected into the tissue, the urea diffuses under a concentration gradient, and the TPs and gelatin chains recombine to trigger the in situ formation of a hydrogel with excellent adhesiveness. Under NIR laser irradiation at 1060 nm for 10 min (1.0 W/cm^2^), results showed that AgPOM fixed in the hydrogel reacted with hydrogen peroxide at the site of infection, enhanced ROS generation, and generated singlet oxygen to kill drug-resistant *S. aureus* in vitro and in vivo, and the hydrogel accelerated wound healing with good biosafety [43].

Peroxide–AgNPs are used for their ability to activate oxidative stress and are considered a promising nanomedicine therapy, especially in chemotherapy, photodynamic therapy, and bacterial disinfection. Silver peroxide nanoparticles (Ag_2_O_2_ NPs) at a concentration of 78 µg/mL capable of controlled release of ROS have been synthesized. The release of bactericidal Ag+ ions and ROS is strictly regulated by the external stimuli of ultrasound (US) and NIR. NIR irradiation at 808 nm (0.7 W/cm^2^) for 10 min showed in vitro and in vivo that Ag_2_O_2_ NPs have improved antibacterial and antibiofilm capacities with a destruction efficiency close to 100% against MRSA, *S. aureus*, *E*. *coli,* and *Pseudomonas aeruginosa*; they also considerably accelerate the closure of cutaneous wounds and show in vivo biocompatibility [47].

ROS are also produced in a system composed of silver bismuth sulfide quantum dots (QD AgBiS_2_). This antibacterial system promotes electrostatic attachment between AgBiS_2_ QDs and negatively charged membrane surfaces of bacterial cells via polyethylenimine (PEI) packaging. The photo-induced antibacterial activity of AgBiS2 QDs (33 nm) (60 μg/mL) was studied against *S. aureus* and *E. coli* under 808 nm laser irradiation (1.6 W/cm^2^) for 10 min. The results revealed a significant reduction in bacterial survival. The antibacterial effect was enhanced by anchoring AgBiS_2_ QDs and inducing a rupture of the outer bacterial membrane by high local photothermal heat and a release of ROS [48]. The same antibacterial mechanistic effect was also reported in a study using graphene quantum dots conjugated to AgNPs (GQD–AgNP) (spherical shape, 40 nm). The conjugation of AgNPs to the surface of GQDs enhances the production of ROS in light-activatable GQDs and the transformation of light energy into hyperthermia with high efficiency. This was determined following the irradiation of bacterial cells of *S. aureus* and *E. coli* treated with GQD–AgNP (0.5 mg/mL) by a laser at 450 nm (14.2 mW/cm^2^) for 10 min [35].

However, the antibacterial activity of the nanoparticles was demonstrated when they were inked in a polymer composed of fluorinated graphene (FG) enriched with oxygen groups (FGO). The FGO–Ag complex exhibits high near-infrared absorption for PTT at 808 nm (2.0 W/cm^2^) for 5 min and exhibits effective antibacterial activity against *E. coli* and *S. aureus* [36]. Graphene oxide was also used in two other studies to produce silver nanoclusters with antibacterial and photothermal activity [49,50].

Ag nanoclusters (~3.57 nm) have been produced directly by the reduction of reduced graphene oxide (rGO) (denoted AgNC/GSH-rGO) using glutathione (GSH). The good electrical conductivity of rGO favors the extremely small particle size of Ag nanoclusters. AgNC/GSH-rGO nanohybrids (300 μg/mL) have also been shown to be effective antibacterial and photothermal agents under the effect of local irradiation at 808 nm (2 W/cm^2^) for 5 min [49]. A hyaluronidase (HAase)-triggered photothermal platform based on AgNPs and graphene oxide (GO) has been designed to be antibacterial. Upon NIR light illumination at 808 nm (1.0 W/cm^2^) for 2 min of *S. aureus* in vitro and in vivo, GO-based nanomaterials (100 μg/mL) increased the temperature locally, resulting in high bacterial mortality in a wound disinfection model. The antibacterial HAase-triggered AgNP (50 nm) release allows AgNPs to be protected by a hyaluronic acid (HA) matrix without affecting normal host cells [50].

In the same context of antibacterial activity and wound healing, a hydrogel model was synthesized based on hyaluronic acid–tyramine (HT) cross-linked by an enzyme loaded with AgNPs, called HTA. Natural antioxidant tannic acids (TA) were used as both reducing and stabilizing agents to easily synthesize TA-capped silver nanoparticles (AgNPs@TA). The incorporation of AgNPs@TA significantly enhanced the antioxidant, antibacterial, photothermal, adhesive, and hemostatic capabilities of the hydrogel, after the application of the hydrogel to the site of infection and irradiation at 808 nm for 10 min (0.92 W/cm^2^). The in vivo results of a cutaneous wound model of mice co-infected with *S. aureus* and *E. coli* showed that the hydrogel (containing 0.4 mg/mL of AgNPs@TA) enhanced antibacterial activity and accelerated wound healing [45]. In addition, a system with a stable plasmonic effect for PTT (silk–GOx–Ag@G, SGA) (70 nm) has been designed based on antibacterial glucose oxidase (GOx) embedded in a network silk film of graphitic Ag nanocapsules (Ag@G) to fabricate a synergistic GOx PTT system. Following in situ identification of bacterial intrinsic signals in a mouse wound model, the plasmonic complex achieves superior synergistic antibacterial effect on infected *E. coli*, *S. aureus*, and MRSA in vivo after 808 nm laser irradiation (5 W/cm^2^), without causing significant biotoxicity [46].

A design concept of a water-soluble fluorinated carbon fiber oxide (FCO)/Ag composite was used as an antimicrobial agent, a highly effective targeting nanocarrier, and photothermal therapy. In vitro and in vivo results revealed the synergistic interactions of lipophilic fluorine and AgNPs that provide highly effective antibacterial activity [51].

With the aim of having a system for delivering an antibacterial metal treatment, hollow mesoporous silica nanospheres (HMSN) were designed and loaded with silver nanoparticles (Ag NPs), vancomycin (Van), and hemin (HAVH) for the elimination of bacteria and abscess treatment. However, this nanocomplex is photosensitive. Following NIR light irradiation (808 nm, 1.0 W/cm) (45 °C) for 10 min at a concentration of 250 µg m/L, hemin is released from the mesopores of HMSN, thus triggering the opening of the pores and the release of preloaded Ag+ and Van. A synergistic photothermo-chemotherapy activity of the nanocomplex was thus obtained to fight against MRSA, both in vitro and in vivo. The results also showed that hemin, which possessed intrinsic ROS scavenging activity, could attenuate the inflammatory response at the treatment site and benefit the wound healing process [52].

### 3.2. Gold Nanoparticles (AuNPs)

The development and application in different fields of multifunctional plasmonic NPs is still ongoing for biomedical applications in combination with photothermal therapy, while inducing local hyperthermia. Multi-tipped Au nanostars (NSs) (spherical shape, 92 nm) with an anisotropic structure were fabricated for PTT for antibacterial applications. The photothermal conversion efficiency of AuNS increases up to 28.75% under 808 nm laser irradiation, and the heat generated was sufficient to kill *S. aureus* [53]. A good biocompatibility and antibacterial activity against *S. aureus* have also been demonstrated by an AuNP-based polymer (50 nm) in vivo and in vitro. This polymer was formed by several antibacterial agents, polyhexamethylene biguanide (PHMB, with bactericidal and anti-biofilm functions), and AuNPs (PHMB@AuNPs) (9 µg/mL). This construct exhibited an excellent synergistic effect to enhance the photothermal bactericidal effect under irradiation of *S. aureus* by an 808 nm NIR laser (2.0 W/cm^2^) for 10 min [54]. The synthesis of an AuNP nanocomplex inked with IgG molecules (spherical shape with diameter of 42 nm) was administered to MRSA cultures, and PTT was performed at 808 nm (2 W/cm^2^) for 10 min. The results of this synthesis indicate that prolonged and selective bacterial death was produced under the effect of local photothermal heat [55].

In addition, AuNPs have also been used as agents for targeting and selective destruction of laser-targeted bacteria of *S. aureus* [56]. This could be achieved by light-absorbing AuNPs conjugated to specific antibodies, by a PT multifunctional microscope/spectrometer system that allows real-time evaluation. AuNPs of different sizes (10, 20, and 40 nm) conjugated to anti-protein A antibodies were used to target the surface of *S. aureus* under irradiation with focused laser pulses (420–570 nm, 12 ns, 0.1–5 J/cm^2^, 100 pulses). However, laser-induced bacterial damage has been observed by light and transmission electron microscopy. By a combined treatment of three approaches, nanoparticles, laser, and PT technique, bacteria were effectively targeted and killed in vitro [56].

In another nanohybrid system, sphere-shaped AuNPs were conjugated to monoclonal antibodies combined with single-walled carbon nanotubes (SWCNTs) to design a nanodrug (SWCNT-GNPs) to detect and selectively eradicate a resistant *Salmonella typhimurium* DT104 strain. A therapy based on this nanodrug (SWCNT-GNPs) combined with targeted PTT using 670 nm light at 2 W/cm^2^ for 15 min, resulted in the selective damage of more than 99% of *Salmonella* DT104 [89].

As in some polymer AgNPs, polydopamine was used to increase the photothermal effect of AuNPs. In this study, bacterial cellulose-based photothermal membranes containing polydopamines anchored to AuNPs (1 mg/mL of dopamine and 30 mmol/L of chloroauric acid) were designed as a model to promote bacterial death and subsequent wound healing. Indeed, the in vitro and in vivo antibacterial efficacy of these nanocomposites was evaluated against *S. aureus*, *E. coli*, and MRSA. The results revealed an antibacterial efficacy of 99% under NIR irradiation at 808 nm (1.0 W/cm^2^), an anti-inflammatory effect, and the regeneration of damaged tissue with a good biocompatibility evaluated by >85% of survival cells [57]. In another study, the photothermal efficiency of polydopamine nanoparticles was demonstrated in a construct of a gold nanostar/hollow polydopamine Janus nanostructure (GNS/HPDA JNPs). The property of this system is essentially the release of nitric oxide (NO) under NIR irradiation. However, the treatment (JNP GNS/HPDA at 808 nm irradiation for 5 min, 125 µg/mL) of bacteria, including *E. coli* (ATCC 25922), *S. aureus* (ATCC 29213), and MRSA (ATCC 43300), showed bactericidal activity against the bacteria tested, through the release of NO. The NO particles could have a synergistic effect by inhibiting the growth and formation of biofilm. The authors were able to explain this effect by the destruction of the bacterial membrane and the degradation of its genetic material or the disturbance of the bacterial metabolism. Interestingly, this nanostructure was able to reduce the bacterial resistance of the MRSA strain via the regulation of the *mecA* gene and certainly of certain genes which code for the proteins responsible for the production and regulation of the efflux pumps functions (SepA and Tet38). In addition, the antibacterial efficacy of the GNS/HPDA JNPs nanostructure combined with PTT was also evaluated in vivo in a rat model with MRSA-infected wounds, and this confirmed the results obtained in vitro with good healing of infected wounds and biocompatibility in vitro and in vivo [58].

A coating of PDA on hydroxyapatite (HAp) embedded with AuNPs (Au-HAp) (4.9 ± 0.9 nm) was developed to be an antibacterial agent that promotes wound healing. PDA@Au-HAp NPs produce hydroxyl (•OH) radicals via the catalysis of a small concentration of H_2_O_2_ which will make bacteria more permissive to thermal irradiation. The antibacterial efficacy of NP PDA@Au-HAp against *E. coli* and *S. aureus* was evaluated at 96.8% and 95.2%, respectively, following irradiation at a controlled photo-induced temperature of 45 °C by a NIR laser at 808 nm (1.0 W/cm^2^) for 10 min at a concentration of 200 µg/mL. In addition, the nanocomposite has been determined in vivo to regulate the expression of genes related to tissue repair to promote angiogenesis and thus accelerate wound healing without causing cytotoxicity [59].

Due to high NIR absorbance, indocyanine green (ICG) has been used in several studies involving photo-theranostic therapy. Porous AuNPs have been synthesized as photothermal agents. AuNPs were loaded with ICG, a common photosensitizer for PDT, to construct a nanostructure exhibiting synchronous PTT and PDT properties under NIR irradiation. The hybrid nanocomposites showed remarkable antibacterial effect against *S. aureus* under 808 nm laser irradiation [61]. Moreover, the photodynamic effect of ICG aggregates was combined with the photothermal effect of Au nanorods (AuNRs) (100 μM) irradiated at 808 nm (1 W/cm^2^) for 1 min to achieve an effective treatment of bacterial infection against *E. coli* (ATCC 8393) and *S. aureus* (ATCC 6538P) [62]. Under resonant laser irradiation, another synthesis of AuNRs becomes very effective in heating extremely useful nano-converters for an antimicrobial concept of AuNRs. NIR illumination at 810 nm (6.3 W/cm^2^) caused an increase in local temperature to ≈50 °C in about 10 min. The results showed that the structure achieved an efficiency of killing *E. coli* bacteria of approximately 2 log CFU (colony forming unit) with good biocompatibility [63]. Light-sensitive AuNRs were prepared by in situ copolymerization by N-isopropylacrylamide (NIPAM), acrylic acid (AA), and N-allylmethylamine (MAA) as monomers on the surface of the AuNRs (150 nm). Ciprofloxacin (CIP) was loaded onto polymer shells of the nanocomposites (i.e., AuNRs) (with a content of 9.8%). The negatively charged nanocomposites are converted into positively charged ones during their accumulation on the infectious sites, which will eventually enhance the penetration into the biofilm and the bacterial adhesion of the AuNRs-CIP. NIR irradiation at 808 nm (1.0 W/cm^2^) for 10 min allows the nanocomposites (9.42 μg/mL) to reach the depth of the tissue to penetrate the MRSA bacterial biofilm, instantly releasing CIP, which will induce a synergistic chemo-photothermal activity. The results have been proven in vitro and in vivo and also against another bacteria: *E. coli*. The cytocompatibility was investigated and it was indicated that a negligible cytotoxicity was displayed in the normal cells [64]. To increase microbial selectivity, AuNRs were grafted with a selective antimicrobial peptide (AMP) called C-At5 (50 nm). This strategy also aims to reduce the cytotoxic effects of nanorods on human cells. However, under the effect of local photothermal irradiation at 808 nm (2.5 W/cm^2^) for 10 min, the AuNRs-C-At5 structure (25 μg/mL) was able to exert an antibacterial effect against the two bacterial strains, *E. coli* (ATCC 25922) and *S. aureus* (ATCC 25923), in vitro. Then, this activity was evaluated in vivo and it was found that the complex based on AuNRs encumbered with a microbial selection peptide facilitates wound healing in a mouse model and promotes better bactericidal activity [87]. The antibacterial photothermal efficacy of AuNRs functionalized with DNA aptamers (Apt@Au NRs) was determined in vitro against a strain of MRSA targeted at PTT at 808 nm (1.1 W/cm^2^) for 2 min [60].

With the aim of eliminating a bacterial biofilm and improving the effect of AuNPs in this context, an interesting nanostructure was constructed by a gold nanocage releasing NO (AuNC@NO) (50 ± 5 nm) following stimulation by NIR irradiation to deliver NO and generate hyperthermia for biofilm removal. However, it has been reported that AuNC@NO was prepared by immobilizing a temperature-sensitive NO donor onto AuNC through thiol–gold interactions. Based on the characteristics of this complex, namely, the release of NO under NIR irradiation to eliminate biofilm, AuNC@NO (100 pM) under NIR irradiation at 808 nm (1 W/cm^2^) for 1 min exhibited improved in vitro bactericidal and antibiofilm efficacy against *E. coli* (ATCC 8393) and *S. aureus* (ATCC 6538P) compared to AuNC alone. Moreover, this efficacy has also been demonstrated in vivo in a subcutaneous biofilm infection model. In vivo results indicated that after 5 min of 0.5 W/cm^2^ NIR irradiation, NO release from AuNC@NO was significantly enhanced, which induced the dispersal of MRSA biofilm, and under the synergistic effect with PTT, planktonic MRSA was killed after losing its biofilm protection. A cytotoxicity test showed good hemocompatibility of the nanocomposite [65]. In the same context, a study was carried out in addition to that cited above to determine the effectiveness of AuNC formulations generated with antibodies (~750 nm) targeting two different lipoproteins of *S. aureus* (SACOL0486 and SACOL0688), antibiotics (ceftaroline, vancomycin, or daptomycin) targeting *S. aureus*, and other (gentamicin) alternative combinations targeting the pathogen *P*. *aeruginosa* (ATCC 27317). The results confirmed that daptomycin-loaded AuNCs conjugated with antibodies targeting the two different lipoproteins of *S. aureus* (SACOL0486 and SACOL0688) effectively kill MRSA in the context of biofilm at 0.4 nM. Furthermore, ceftaroline- and vancomycin-loaded AuNCs conjugated to anti-Spa antibodies were found to exhibit reduced efficacy compared to daptomycin-loaded AuNCs conjugated to the same antibody. By contrast, gentamicin-loaded AuNCs conjugated to an antibody targeting a conserved outer membrane protein were highly effective against *P*. *aeruginosa* biofilms, and these results were obtained under NIR laser irradiation conditions at 808 nm (0.8 W/cm^2^) for 10 min [67].

Another model of NO nanogenerator has been developed for biofilm eradication. PDG@Au-NO/PBAM is composed of the heat-sensitive NO-donating conjugated AuNPs on cationic poly (dopamine-co-glucosamine) (PDG@Au–NO) nanoparticles, and an anionic copolymer of phenylboronic acid and acryloylmorpholine (PBAM), which makes up the shell. However, according to the authors, PDG@Au–NO/PBAM seems to be a good generator of NO; when it reaches the biofilm, its surface charge would be positive after the dissociation of the shell and the exposure of the cationic nucleus, which will allow it to infiltrate and accumulate in the depth of the biofilm. At a concentration of 150 μg/mL and under NIR irradiation at 808 nm, 1.0 W/cm^2^ for 10 min, PDG@Au–NO/PBAM could sustainably generate NO to disrupt biofilm integrity under hyperthermia (54 °C). This serves to effectively eradicate the biofilm of resistant bacteria, MRSA (ATCC BAA-40) and TREC (ATCC ER2738). Moreover, the in vitro cytotoxicity of the PDG@Au–NO/PBAM nanogenerator with mouse embryonic fibroblast cells (NIH3T3) showed that PDG@Au–NO/PBAM at a concentration of 150 μg/mL could promote cell survival and proliferation through protection against apoptosis [66].

Another study demonstrated that the antibacterial therapeutic activity of chitosan-coated AuNPs was enhanced by curcumin (Cur) (nearly spherical shape; 16.21 nm and 20.89 nm). Chitosan as an outer layer covered with AuNPs could improve the dispersibility of Cur. The AuNPs prevent the rapid photobleaching of curcumin causing PTT, thus ensuring the yield of singlet oxygen under irradiation, and improving the electrostatic bonding with the cell membrane of bacteria, which will thus promote the bactericidal effect of this nanostructure. Under irradiation at the two wavelengths of 405 nm and 808 nm for 5 min, the AuNPs/CS-Cur kills *E. coli*, *P. aeruginosa*, *B. subtilis*, and *S. aureus* via the production of ROS at 4 μM. In addition, the complex significantly enhanced the synergistic PTT/PDT photoinactivation against *S. aureus* and *E. coli* without cytotoxicity [78].

Gold nanoplates (AuNPT) (triangular plates, 132 nm) have been shown to exhibit significant photothermal conversion efficiency (68.5%) and peroxidase-like activity when accompanied by H_2_O_2_. The combination of the very low concentration of H_2_O_2_ (0.1 mM) and the low dose of AuNPT (50 μg/mL) with laser irradiation at 808 nm (1 W/cm^2^, 3 min) shows excellent synergistic antibacterial ability, and this has been demonstrated in vivo in a patient infected with MRSA [33].

Furthermore, the photothermal-sensitive sodium nitroprusside (SNP) inside the MIL-101-NH_2_ (MOF) gold–maleimide (SNP@MOF@Au–Mal) will release NO and generate ROS in situ under NIR irradiation at 808 nm (1.5 W/cm^2^) for 10 min to achieve a targeted synergistic antibacterial effect in vivo by maleimide which can recognize and bind to *P*. *aeruginosa* T4P pilus (97.7% bacterial cgar re-education). Additionally, the nanogenerator has been shown to promote the secretion of growth factors, which play a key role in regulating inflammation and inducing angiogenesis [68], and to increase the penetration of antibacterial agents into the biofilm, a nanoswimmer based on AuNPs and vancomycin (Van) driven by NIR light (HSMV) was designed. Under 650 nm laser NIR light irradiation at 1.5 W/cm^2^ for 10 min, HSMV performs efficient self-propulsion and penetrates the biofilm within 5 min due to the photothermal conversion of asymmetrically distributed AuNPs. The Van’s localized (∼45 °C) heat-triggered thermal release leads to an effective combination of photothermal therapy and chemotherapy in one system. Active movement of HSMV increases the effective distance of the PTT and also improves the therapeutic index of the antibiotic, resulting in a superior biofilm removal rate of *S. aureus* (>90%) in vitro and in vivo [69]. Another study demonstrated that the complex of vancomycin-immobilized AuNPs (Au@Van NPs) of polygonal shapes exhibits both antibacterial capacity and photothermal competence. At a temperature of 15 °C under NIR laser irradiation at 808 nm for 5 min, the sample containing Au@Van NPs exerted antibacterial activity against vancomycin-resistant Enterococci (VRE) [70].

Nanocomposites based on lanthanide-doped multifunctional upconverting nanoparticles (UCNPs) were sandwiched with gold (gold (Au^1^)–UCNP–gold (Au^2^)) (10 ± 1 nm and 100 ± 2 nm) so that the polymer had the ability to exhibit an effect at ultra-low thresholds under continuous-wave (CW) laser excitation. Nanocomposites subjected (Au^1^ (100 nm), Au^2^ (10 nm)) to stretchable systems at 980 nm (0.2 kW/cm^2^) for 20 min via a light-trapping effect demonstrated antibacterial activity by PTT against *E. coli* (BRBC 12438) and *S. aureus* (10780) [71]. For the theranostic purpose of bacterial infection, a multifunctional plasmonic gold chip was constructed. The chip exhibits high bacterial capture efficiency, plasmon-enhanced fluorescence (PEF), and surface-enhanced Raman scattering (SERS), and can act as a highly sensitive sensor for dual-mode imaging and the detection of bacteria (up to 10^2^ CFU/mL) with good reliability and accuracy. The developed test can distinguish *S. aureus* (i.e., cocci) from *E. coli* (i.e., bacilli). In addition, this chip exhibits antibacterial activity via its PTT property in vitro and in vivo [72].

Nanodrugs composed of stable daptomycin–gold nanoflowers (Dap-Au_n_NF) (round shape, 20 nm) have been used as antibacterial agents. The photothermal conversion of this nano compound is 40% efficient. Dap-Au6NF inhibited the growth of *E. coli* (52%) and *S. aureus* (64%) under NIR radiation at 808 nm (1.75 W/cm^2^) for 10 min at 200 μM. At this concentration, the nanodrug exhibited good biocompatibility [73]. Additionally, vancomycin-modified gold nanostars have been constructed to kill antibiotic-resistant bacteria. After the antibacterial experiments, the AuNSs@Van nanodrug was able to kill MRSA under NIR irradiation at 808 nm (2.5 W/cm^2^) for 10 min in vitro and in vivo [74]. In a theranostic strategy, vancomycin and the compound E-cyclooct-4-enyl-2,5-dioxo-1-pyrrolidinyl carbonate (TCO-NHS) were used for bacterial targeting. This compound was coupled to aggregates of AuNPs to target bacterial destruction. Plasmonic coupling between adjacent AuNPs showed a strong “hotspot” effect, allowing efficient SERS imaging of bacterial pathogens. More importantly, in situ aggregation of AuNPs showed strong NIR adsorption at 808 nm (2 W/cm^2^) for 5 min allowing enhanced photodestructive activity against *Bacillus subtilis* (ATCC 6633), *S. aureus* (ATCC 700698), *Enterococcus faecalis* (ATCC 29212), and *E. coli* (ATCC 53868) bacteria. The authors suggested that this bioorthogonal theranostic strategy could have potential applications in the treatment of bacterial infections [75].

Nanoclusters have been developed by functionalized gold deoxyribonuclease (DNase) (AuNCs) (DNase-AuNCs). This nanocluster is designed to kill *E. coli* and *S. aureus* in particular by dispersing the surrounding biofilms. Under the effect of DNase, the DNase-AuNC nanocompound could decompose the extracellular polymer substance matrix to expose the bacteria to PTT and PDT. However, following irradiation at 808 nm (2 W/cm^2^) for 10 min, the combination of enzymolysis, PDT, and PTT can effectively remove biofilms with a scatter rate of up to 80% and kill about 90% of the protected bacteria. This treatment is intended primarily to eliminate bacterial biofilm from orthodontic devices [76].

Gold nanoshells coated with chitosan thiol (TC-AuNS) (120 nm) as an antibacterial agent combined with the effect of PTT have been developed against *S. aureus*, *E. coli*, and *P*. *aeruginosa*. A concentration of 115 μg/mL of the TC-AuNS nanocompound was shown to have a significant antibacterial effect against the three bacterial strains tested after irradiation at 808 nm (0.95 W/cm^2^) for 5 min [77].

Toluidine blue O (TBO) and AuNPs were used as ROS generators in the presence of light. The complex exerts bactericidal activity against *E. coli* and *Bacillus cereus* after exposure to green light at 530 nm for 5 min [79].

Bacteriophage M13 incubated with colloidal Au was studied as a nanoscaffold (161 ± 33 nm) for plasmonic bactericidal agents. The affinity of the Au layer on the surface of viral scaffolds was assessed by electron microscopy.

The bactericidal and phototheric activity of the complex was achieved following laser irradiation at 532 nm with a variety of powers and exposure times (0, 100, 200, and 300 mW/cm^2^) for 20 min. The results demonstrated an antibacterial activity evaluated at 64% of *E. coli* (K12 ER2738) [81]. The use of bacteriophages (phages) for antibacterial therapy is increasing, and in another study, phages were bioconjugated to create “phanorods” (Phage-AuNPs) that can target bacteria. Under the effect of phanorod irradiation at 808-nm (3.0 W/cm^2^) for 10 min, the phanorods were effective in killing *E. coli*, *Vibrio cholerae*, and also eliminating a biofilm of *P*. *aeruginosa* cultured on epithelial cell culture. In addition, the phanorods are inactivated during the irradiation, to prevent their replication and therefore their evolution during the therapy period. The phanorod methodology developed in this study is best suited to the treatment of directly accessible tissues or surfaces, essentially in the case of wound infections or the colonization of medical devices [86].

Another approach aimed at the fabrication of porous silicon nanopillars decorated with AuNPs (∼250 nm) to improve their photothermal conversion properties. When these nanostructures (0.5 mg/mL) were irradiated with a laser at 808 nm (1.25 W/cm^2^) for 10 min, a reduction in bacterial viability of up to 99% was demonstrated in vitro against *E. coli* (ATCC 25922) and *S. aureus* (ATCC 29213) [82].

AuNPs (14 nm) were readily prepared by surface modification with pH-sensitive mixed charged zwitterionic self-assembled monolayers consisting of weak electrolyte 11-mercaptundecanoic acid (HS-C10-COOH) and strong electrolyte (10-mercaptodecyl) trimethylammonium bromide (HS-C10-N4). These modified NPs exhibited a rapid pH-sensitive transition from negative to positive charge, which allowed the AuNPs to disperse well in healthy tissues (pH ∼7.4), while rapidly exhibiting strong adhesion to the surfaces of bacteria (charged negatively) in MRSA biofilm (pH ~5.5). Under the effect of photothermal irradiation at 808 nm (0.91 W/cm^2^) for 10 min, the modified AuNPs exert anti-MRSA antibacterial activity while eliminating the biofilm. At the same time, this therapy was targeted at infected tissues with low cytotoxicity [83].

Spherical AuNPs (50 nm in diameter) coated with polyethylene glycol were synthesized and added to a culture of *E. coli*. Under irradiation at 532 nm (60 mW/cm^2^) for 5 min, a bactericidal effect with 99% inhibition of bacterial growth was observed [84].

Gold nanobipyramids (AuNBP) (82.24 ± 3.34 and 24.90 ± 1.75 nm) containing a (111) plane have been also used as photothermal agents for non-invasive photothermal therapy. These AuNBPs revealed photothermal capacity and reversibility of the laser response under NIR laser irradiation at 808 nm. AuNBPs (25 µg/mL) showed higher efficacy than AuNRs under 808 nm (1.0 W/cm^2^) NIR laser irradiation for 10 min against *E. coli* without cytotoxicity [85].

In another strategy that serves to increase the photothermal capacity of AuNPs, the latter were coated with composite carbon dots (N,S-CDs) to form Au@CDs (Au@CD) (∼530 nm). Then, the Au@CDs were embedded in a polyvinyl alcohol membrane by electrospinning. The antimicrobial activity and wound healing capacity of the dressing was evaluated in vitro against *S. aureus* (ATCC 25923) and *E. coli* (ATCC 25922) under 808 nm irradiation (3 W/cm^2^) laser NIR irradiation for 10 min. The results showed that the antibacterial activity of the nanocomplex is dependent on the Au@CD (100 mg/mL) content under the effect of irradiation, thus achieving a result of 100% inactivation of the two pathogens. Then, the authors evaluated the photothermal antibacterial efficacy and healing ability of the Au@CD membrane in vivo. The experiment was carried out in male Kunming mice infected with *S. aureus* (ATCC 25923) with the Au@CD membrane applied and irradiated for 5 min. The PTT of the treated wound area reached ∼50 °C. The Au@CD membrane reacted effectively to kill bacteria (99%) under NIR irradiation without damaging the surrounding tissue (70% of cell viability) [88].

### 3.3. Bimetallic Nanocomponents Based on Au or Ag

The evidence that has demonstrated the antibacterial efficacy of bimetallic nanocomponents based on Au or Ag is presented in Table 2.

A nanocapsule favoring the release of kanamycin under a photothermal effect has been developed in a hybrid material combined with and SiO_2_ which will facilitate the administration of the drug coated with AuNRs to increase the efficacy of PTT. After the treatment of a culture of *E. coli* BL21 strain, irradiation with an 785 nm emitting diode laser (120 mW/cm^2^) for 20 min showed a bactericidal effect of the nanocapsule [92].

In the context of the treatment of periodontal diseases, a bimetallic nanoplatform of Au/Ag nanoparticles has been linked to procyanidins (PC) (122 nm). The antibacterial activity of this Au/Ag-PC nanocomposite is initiated by PTT at 150 μg/mL with low cytotoxicity (88%). The presence of PC could regulate host immunity by eliminating intracellular ROS and decreasing the local proinflammatory reaction. Under the effect of irradiation at 808 nm by an NIR laser for 10 min, Au/Ag-PC exerted an excellent bactericidal effect against the bacterium *Porphyromonas gingivalis* (ATCC 33277) and promoted tissue repair in animal models with periodontal inflammation [93]. A bimetallic complex consisting of AuNPs and AgNPs was tested for its ability to eliminate a bacterial biofilm formed by *B*. *subtilis* under irradiation for 5 min at 808 nm (1 W/cm^2^). This antibacterial activity was confirmed on biofilms formed by *E coli* (K-12 strain, WT) and *P*. *aeruginosa* (ATCC 27853) but not on that formed by *S*. *epidermidis* (ATCC 12228). The authors were able to suggest that this result could be due to the poor metallic distribution in the biofilm of *S*. *epidermidis* as well as to the short period of irradiation [94]. Another study was able to achieve a synthesis of Au/Ag bimetallic nanoparticles (size 23 nm) without stabilizers or surfactants. Additionally, these Au/Ag bimetallic nanoparticles have applications for PTT. This activity was validated during the treatment of bacterial cultures of *S. aureus* with the complex of bimetallic nanoparticles (50 μg/mL) under laser irradiation at 808 nm (2 W/cm^2^) for 5 min. Significant antibacterial activity has been revealed in vitro and in vivo in animal models with *E. coli* skin infections without cytotoxicity. The successful treatment of the wound was carried out in 96% of the group of animals that received the Au/Ag NP and PTT treatment [95]. A hybrid gold–silver nanocage (Au/Ag NCs) is designed to be conjugated with an antimicrobial peptide (AMP) and hyaluronic acid (HA) via Au–S bonding and electrostatic adsorption, respectively. The HA–peptide (P) (Au/Ag) exhibits small size (128 nm), high photothermal conversion efficiency, and good stability. Under NIR irradiation at 808 nm, 1.0 W/cm^2^, 10 min, the HA-P (Au/Ag) nanocage (12 μg/mL) effectively kills multidrug-resistant bacteria like *Acinetobacter baumannii* (MDR-AB) by disrupting their inner and outer membrane in vitro. Furthermore, this activity has been demonstrated in an in vivo model of pneumonia caused by MDR-AB. HA-P (Au/Ag) treatment reduced the number of bacterial colonies and inflammation in the lung tissues, and restored the structure of the pulmonary alveoli with a restoration of life in infected mice, with a cell viability higher than 95% [96]. For an antibacterial treatment strategy, a novel targeting delivery nanosystem was designed based on an Au/Ag NC that could efficiently interact with bacterial recognition receptors found on macrophage membranes (coating *S. aureus* pretreated macrophage membranes onto AuAgNCs) (Sa-M-AuAgNC). Under NIR laser irradiation at 808 nm (1.0 W/cm^2^) for 5 min the Sa-M-AuAgNC nanosystem (200 µg/mL) is efficiently delivered and retained at the site of infection to exert bactericidal activity against *E. coli* (ATCC 43888) or *S. aureus* (ATCC BAA-1721); the system also exhibits a significantly prolonged blood circulation time when it comes to local or systemic injection [97].

In another structure, Au/Ag nanoshell nanosystems (Au/AgNSs) (49.5 ± 4.5 nm) were conjugated with 3,3′-diethylthiatricarbocyanine (DTTC) iodide and fabricated with strong NIR laser response due to surface plasmon resonance (SPR). The photothermal eradication of multidrug-resistant bacteria by this nanosystem (18.7 μg/mL) was evaluated against the four strains *E. coli* (ATCC 25922), *E. coli* (ESBL), *S. aureus* (ATCC 6538), and MRSA. Irradiation with a laser at 808-nm (1.0 W/cm^2^) for 10 min stimulated the release of Ag^+^ ions which served to kill the bacteria tested in vitro. In addition, the bactericidal effect of these nanostructures was evaluated in vivo in a mouse model infected with chronic MRSA, and PTT mediated by the release of Au/AgNSs gel leads to synergistic healing with no evident cytotoxicity to human cells [106].

PDA-coated AuNRs were charged with a Ag+ ion and chitosan glycol (GCS) conjugation labelled with Cy5-SE fluorescent agent. In MRSA and *E. coli* bacterial cultures lent to the Ag +-GCS-PDA@GNRs complex, irradiation by PTT at 808 nm (0.5 W/cm^2^) for 7 min allowed the release of Ag+ ions which served the electrostatic targeting of negatively charged bacteria and bacterial membrane damage, thereby causing increased permeability and reduced heat resistance of the cell membrane [113].

AuNPs or AgNPs were also grafted to other metal particles separately to increase the synergistic antibacterial activity of the two metals.

Magnetic nickel oxide (NiO) NPs were assembled with vancomycin (Van)-modified particles. The NiO NPs@AuNPs@Van (NAV) (193.08 ± 1.61 nm) nanocomposite was synthesized for the selective elimination of MRSA. Under NIR irradiation at 808 nm (1.8 W/cm^2^) for 10 min, Van-mediated AuNPs self-aggregate (125 μg/mL) on the surface of MRSA, and, thanks to the magnetic NiO nanoparticles, a photothermal effect is generated to exert antibacterial efficacy which was able to kill in situ more than 99.6% of the MRSA tested in vitro as well as in vivo, with good blood compatibility (hemolysis rate 0.87%) [98].

In the context of wound healing, Au nanoclusters have also been used to form an organo-metallic zirconium-based porphyrin complex (AuNCs@PCN) (Au 2–5 nm). Under NIR laser irradiation at 56.2 °C, using an 808 nm wavelength laser (1.0 W/cm^2^) for 10 min, AuNCs@PCN can produce ROS acting on bacteria like *S. aureus* (ATCC 25923), MRSA (ATCC 43300), *E. coli* (ATCC 25922), and *Ampr E*. *coli* (ATCC 35218). This organometallic complex could alter the membrane of the bacteria tested when they were irradiated. Indeed, the antibacterial effect of AuNCs@PCN was reproduced in diabetic rats treated with AuNCs@PCN. The wound was reduced by 2.7% in 21 days. No obvious cell cytotoxicity (90% viability) was observed [100].

Recently, hollow silver–gold alloy nanoparticles immobilized with the photosensitising molecule Ce6 (Ag@Au-Ce6 NPs) integrated with PTT at 808 nm (800 mW/cm^2^ for 5 min) and a 660 nm laser (200 mW/cm^2^ for 5 min) were designed to accelerate wound healing in the context of infection by *S. aureus* (ATCC 25923) or *E. coli* (ATCC 25922). As a result of the heat effect provided by an NIR laser, Ag@Au-Ce6 NPs at 0.25 nM could effectively kill the free and colonized bacteria on the surface of the injured skin via the release of ROS, thereby promoting the vascularization of the skin epithelium and wound healing [101].

Another bimetallic nanoparticle model has been developed by synthesizing non-spherical α-Fe_2_O_3_@Au/PDA core/shell nanoparticles (400 nm) with tunable shapes. NIR phototherapy-induced bactericidal performance at 808 nm (2.0 W/cm^2^) for 5 min indicates that α-Fe_2_O_3_@ Au/PDA (50 μg/mL) hybrid particles with tunable non-spherical shapes possess significant antibacterial effects against *E. coli* and *S. aureus* and excellent biocompatibility [102].

BP nanosheets containing AuNPs were assembled in situ with zinc oxide (ZnO) nanoparticles to form a NIR-light-sensitive nanoplatform. Subsequently, the antibacterial activities of the resulting Au-ZnO-BP (50 µg/mL) was observed against non-resistant *S. aureus* species and MRSA after irradiation at 808 nm (2.5 W/cm^2^) for 5 min [104].

Gold–platinum nanodots (AuPtNDs) (2.5 nm) with potent peroxidase-like activity also have excellent photothermal conversion efficiency (50.53%) and broad-spectrum in vitro antibacterial activity against *E. coli* (97.1%) and *S. aureus* (99.3%), when the nanocomplex (80 μg/mL) was irradiated at 808 nm (1 W/cm^2^) for 15 min. AuPtNDs further showed in in vivo experiments that nanodots can effectively promote the healing of bacterial infection wounds and do not damage the normal cells [108].

AuNRs photoexcited by NIR light were immobilized on the titanium (Ti) surface by an electrostatic surface self-assembly technique. In vitro study revealed that the prepared surface modified by AuNRs (rod shape, 11 ± 2  nm) exhibits antibacterial activity against four kinds of bacteria: *E. coli* (ATCC 25922), *P*. *aeruginosa* (ATCC 27853), *S. aureus* (ATCC 25923), and *S*. *epidermidis* (ATCC 12228) under 808 nm irradiation (0.5 W/cm^2^) for 20 min. The Ti AuNRs also exhibit low cytotoxicity [109].

Oxidized alginate (ADA) and catechol-modified gelatin (Gel-Cat) were selected as polymer backbones to construct an antibacterial hydrogel dressing that aids the healing of infected wounds. Both compounds were introduced to polydopamine-decorated AgNPs (PDA@Ag NPs) (104.09 ± 23.88 μm). As we have seen in the studies cited above, PDA increases the photothermal activity of the nanostructure. In vitro experiments show that the hydrogel (150 mg/mL) has a bactericidal capacity against *S. aureus* and *E. coli* under PTT laser irradiation at 808 nm (1.3 W/cm^2^) for 10 min. These results were confirmed in vivo, and it was also observed that the hydrogel helped to activate angiogenesis [99]. Another injectable hydrogel was synthesized by incorporating bioactive glass nanoparticles conjugated to Ag_2_S nanodots (BGN-Fe-Ag_2_S) in a biodegradable solution of poly (ethylene glycol) double acrylates (PEGDA) and (2,2′-azobis [2-(2-imidazolin-2-yl) propane]-dihydrochlorid) (AIPH) to inhibit bacterial growth and promote wound healing in the context of tumor surgery. However, under laser irradiation at 808 nm (1.0 W/cm^2^) for 10 min, the Ag_2_S nanodot-mediated photothermal effect would trigger the decomposition of AIPH, generating alkyl radicals to initiate the gelation of PEGDA. The gelatinized hydrogel exerted a bactericidal activity in situ against *S. aureus*. Additionally, the hydrogel significantly accelerated wound healing with more skin appendages in the full-thickness skin wound model due to bioactive glass hydrolysis [103].

Silver–bismuth nanoparticles supported by mesoporous silica (Ag-Bi@SiO2 NPs) (15.13 nm) are constructed for synergistic antibacterial therapy. In vitro experiments indicate that hyperthermia from PTT irradiation at 808 nm (1 W/cm^2^) for 15 min of 100 µg/mL of the Ag-Bi @ SiO_2_ NPs nanostructure disrupts cellular integrity and accelerates the release of Ag ions, further exhibiting excellent antibacterial performance against MRSA and good biocompatibility [105]. Likewise, nano-silver and zinc oxide (NP Ag and ZnO) deposited on the surface of reduced GO (rGO) exhibited antibacterial activity against *E. coli* K12 following rapid microwave irradiation [107].

A MOF/Ag-derived nanocomposite (110 nm) with effective metal ion release capability and a photothermal conversion effect has been developed. The MOF is composed of a metallic zinc and a graphite-like carbon framework introduced to AgNPs. Upon NIR irradiation at 808 nm (3 W/cm^2^) for 10 min, Zn^2+^ and Ag+ ions are released to target the bacterial structure. Antibacterial experiments against *E. coli* and *S. aureus* have revealed that a low dose of the nanocomposite (0.16 mg/mL) is capable of having an antibacterial efficacy, evaluated at 100%. In vivo evaluation has indicated that the nanocomposites can achieve rapid and safe sterilization of infected wounds with a low cytotoxicity [110].

In a model composed of several metals, AgNPs were linked to a compound based on ZIF-8 and the whole construct was encapsulated by polydopamine and the photothermal agent ICG to constitute ICG@ZIF-8/PDA/Ag (480 ± 16 nm). An experiment showed that after 20 min of irradiation of 100 μg/mL of this compound at 808 nm at 1.5 W/cm^2^, ICG@ZIF-8/PDA/Ag exhibited 100% bactericidal effects against *E. coli* and *S. aureus*, resulting from both hyperthermia from ICG and PDA, and chemical toxicity from the released Ag and Zn ions. When the bacterial incubation period was extended to 12 h, the minimum bactericidal concentration (MBC) of ICG@ZIF-8/PDA/Ag was reduced to 6.25 μg/mL, and this extremely low MBC was due to the long-term chemo-photothermal combinatorial effect induced by NIR irradiation. Additionally, the composites successfully promoted healing of *S. aureus*-infected wounds in mice with excellent biocompatibility [111].

Another hybrid nanosystem based on ZIF and Ag (Ag_2_S@ZIF-Van NS) (53.4 nm) was constructed by one-step self-assembly of Zn^2+^ vancomycin (Van) and Ag_2_S quantum dots (QD). The utility of Ag_2_S@ZIF-Van was studied in vivo under acidic pH/photothermal dual stimulation. The infected site was treated by being sprayed with 200 μg/mL of Ag_2_S@ZIF-Van, after which the temperature of the wound infection site can rapidly rise to over 50 °C after 10 min of NIR laser irradiation, revealing it to be a potentially effective photothermal therapy. A blood compatibility evaluation demonstrated a 10% hemolysis rate at 200 μg/mL and an absence of inflammatory lesion tissue damage.

In vitro and in vivo experiments have demonstrated that the Ag_2_S@ZIF-Van system is a promising multifunctional tool for enabling accurate diagnosis of bacterial inflammation and treatment of bacterial wound infection [112].

## 4. Ag- and/or Au-Based Compounds Used in Photoacoustic Therapy (PTAT)

Optical techniques for imaging the sites of microbial infections in vivo are limited by the shallow imaging depth and highly light-scattering tissue. Photoacoustic therapy (PTAT) has been used as a theranostic method in the case of bacterial and fungal infections. In this part, we discuss metallic nanostructures based on Ag and/or AuNPs for antimicrobial theranostic applications (Table 3).

In the treatment of fungal keratitis, ethylenediaminetetraacetic acid (EDTA)-modified Ag and Cu_2_O hybrid heterojunction nanoparticles (AgCuE NPs, size 250 nm) were synthesized to disrupt the cell wall of *C*. *albicans*. AgCuE NPs completely inhibited fungal growth at a concentration of 20 μg/mL under laser irradiation at 808 nm (0.25 W/cm^2^) for 5 min. Additionally, AgCuE NP-based gel formulations were applied topically to the cornea with the in vivo fungal keratitis model. Optical coherence tomography and PA imaging were used to monitor nanogel retention and assess the effect of keratitis treatment on infected murine corneas. Optical coherence tomography can monitor the effect of corneal AgCuE NP treatment by determining the thickness and edema. This therapy suggests that the AgCuE NP gel is an effective and safe antifungal strategy for fungal keratitis with favorable prognosis and potential for clinical translation. It also shows good biosafety and no obvious ophthalmic and systemic side effects, with cell viability evaluation greater than 80% at a concentration of 80 μg/mL [24].

A theranostic imaging strategy was developed to monitor an antibacterial therapy based on glucose polymer (GP)-modified AuNPs (150 nm) (1.0 mg/mL). The bacteria must ingest these nanoparticles, taking up the GP as a carbon source through the ATP-binding cassette transporter (ABC) pathway. In infection models caused by *E. coli* (ATCC 11303), *S. aureus*, *Micrococcus luteus* (BNCC 102589), and *P*. *aeruginosa* (BNCC 125486), aggregates of bacterial cells were distinguished using PA imaging following laser irradiations at 405 nm (1.0 W/cm^2^) for 25 min, at 660 nm (12 mW/cm^2^) for 5 min, and at 808 nm (1.0 W/cm^2^) for 5 min in vivo. The aggregates display approximately 15.2-fold improvement in photoacoustic signals and approximately 3.0-fold improvement in antibacterial rate compared to non-aggregated counterparts. The cell viability was evaluated at up to 80% when the cells were incubated with the GP-modified AuNPs [116].

As we have already mentioned (see above [38]), the Ag@BP nanostructure has antibacterial activity; a study revealed that a BP vesicle coupled to Ag+ (BP Ve-Ag^+^) (10 nm) gains a second capacity for PA imaging at a near-infrared window (NIR-II). Under PTT irradiation at 660 nm (150 mW/cm^2^), the BP Ve-Ag^+^ complex could release Ag^+^ ions at the site of infection with a vesicle assembly. This process is guided by NIR-II PA bioimaging. BP Ve-Ag+ can contrapuntally kill *S. aureus* pathogenic bacteria and accelerate wound healing monitored by NIR-II PA imaging. The nanocomposite exhibits moderate cytotoxicity [118].

A photoacoustic contrast agent was developed based on functional peptide-modified spherical gold nanoparticles (AuNPs@P1) (26.4 ± 9.7 nm) with the sequence CLVFFAEDPLGVRGRVRSAPSSS, which confers the specific binding of these nanoparticles to the *S. aureus* cell surface. These NPs could be surface peptides specifically adapted by an overexpressed bacterial enzyme inducing the in situ aggregation of AuNPs (200 µg/mL). Targeting under 710 nm irradiation of *S. aureus* (ATCC 6538) and *E. coli* (ATCC 25922) by these modified NPs showed antibacterial efficacy and low cytotoxicity. Similarly, this activity was shown in vivo by active targeting and aggregation induced by in situ stimuli, which contributed to increasing the accumulation of NPs in the infected site. This dynamic aggregation results in significantly sensitive and specific photoacoustic signals for bioimaging of bacterial infection in vivo [119]. Additionally, other evidence has shown that AuNPs (40 nm) conjugated to antibodies specific for *S. aureus* peptidoglycan were incubated with suspensions of MRSA and methicillin-susceptible *S. aureus* (MSSA). The bacterial suspensions were then exposed to pulsed laser irradiation of 8 ns at a wavelength of 532 nm and fluences ranging from 1 to 5 J/cm^2^. PTT-activated nanoparticle treatment reduced the surviving population of both MSSA and MRSA strains. Significant decreases in bacterial viability occurred when the laser fluence exceeded 1 J/cm^2^ [122].

In a very interesting approach to follow a treatment intended against *Helicobacter pylori* in the stomach, polyclonal *Helicobacter pylori* antibodies, resulting in pH-sensitive Au nanostars@H (80 ± 7 nm), were designed. These nanoprobes (AuNS@Ab) (0.4 mg/mL) have been used for theranostics of *H. pylori* in vivo. PA imaging confirmed that the prepared AuNS@Ab could actively target *H. pylori* in the stomach. AuNS@Ab nanoprobes could kill *H. pylori* in vivo in animal models under 790 nm NIR laser irradiation (1.0 W/cm^2^) for 8 min; all AuNS@Ab nanoprobes could be excreted out of the intestine in the 7 days after oral administration with excellent biocompatibility determined in gastric epithelial cells [120].

Furthermore, it is important to make an early diagnosis of a bacterial infection in the cerebrospinal fluid (CSF). For this reason, researchers have developed new CSF tests based on in vivo PA flow cytometry and photothermal scanning cytometry using AuNRs (size 100 nm) in the CSF of tumor-bearing mice infected with *S. aureus* (ATCC 49230). With this technology, leukocytes, erythrocytes, melanoma cells, and bacteria were detected and counted [123]. For the same purpose of ultrasensitive diagnosis of bacterial infections, in vivo PA flow cytometry has been used for the time-resolved detection of circulating absorbent objects, either without labelling or with nanoparticles as markers for PA. PA flow cytometry’s capability with 830 nm, 100 mJ/cm^2^ tunable NIR pulsed lasers has been demonstrated for real-time monitoring of AuNRs, S. *aureus* (ATCC 49230), *E. coli* K12 tagged with carbon nanotubes (CNTs) (70 nm) (0.5 mg/mL), and Lymphazurin contrast dye in mouse and rat ear and mesenteric microvessels. PA flow cytometry showed an in vivo sensitivity threshold for detecting AuNPs in a volume of irradiated *S. aureus* without cytotoxicity [125].

Bicolored, multilayered magnetic AuNPs with giant PA and PT contrast enhancements were functionalized with an antibody cocktail for the molecular targeting of *S. aureus* surface-associated markers such as protein A and lipoprotein. This method could be applicable for the diagnosis and treatment of bacteria circulating in the bloodstream using intrinsic near-infrared absorption of endogenous carotenoids with nonlinear PA and PT contrast enhancement. In vivo, this advance enabled the ultra-sensitive detection of circulating bacterial cells and their irradiation under the effect of PTT at 850 nm; 0.8 mJ/cm^2^. Therapeutic efficacy was plotted in real time by PTAT [124].

Au/Ag hybrid nanoparticles obtained by coating AuNR with Ag (Au/AgNRs) (104.9 nm) were used as a photoacoustic signature agent. PA contrast is recovered simultaneously when Ag is released, and this PA signal provides non-invasive monitoring of the localized release of Ag^+^ ions. Following an in vivo treatment of mice infected with MRSA and *E. coli*, the released Ag^+^ ions showed strong bactericidal efficacy similar to equivalent free Ag^+^ ions, at an Ag^+^ concentration of 32 μM and 8 μM of MRSA and *E. coli*, respectively. Ag^+^ ions cause low toxicity to mammalian cells [121].

Recently, dual plasmonic AuNR-SiO_2_-Cu_7_S_4_ antibacterial nanomotors have been developed with a Janus configuration predicted by the proliferation of copper-rich Cu_7_S_4_ nanocrystals at a high curvature site of AuNRs, with a diameter of ∼95 nm in length and ∼18 nm. Upon exposure to NR light at 1064 nm (0.75 W/cm^2^) for 5 min at a concentration of 100 μg/mL, a local photothermal field is formed near the AuNR–Cu_7_S_4_ interface, thus resulting in improved photothermal performance and the antibacterial activity of photocatalytic ROS generation. In vivo treatment performed by AuNR–SiO_2_–Cu_7_S_4_ synchronous autonomic movement triggered by NIR light and synergistic photothermal/photocatalytic/photoacoustic antibacterial nanomotors improves transdermal penetration and effectively treats MRSA (ATCC43300) infections at higher efficacy by 98%. In addition, excellent biosecurity has been assessed. Cellular cytotoxicity tests demonstrated the good biocompatibility of the nanomotors [126].

An effective theranostic anti-biofilm agent PTNP based on Au@Au core–cage structures has been developed allowing rapid photoablation and biofilm disruption. In vitro and in vivo treatment with 3.75 mg of PTNP accompanied by NIR irradiation at 808 nm (2 W/cm^2^) for 10 min against *Streptococcus mutans*, UA159 (ATCC 700610), and MRSA (ATCC43300) demonstrates the efficacy of photothermal conversion, a capacity for imaging of biofilm bacteria, and their rapid elimination. This structure is non-toxic and biocompatible [127].

## 5. Conclusions

We have seen in this review that researchers are working hard to develop strategies to tackle antibiotic resistance. Chemotherapy coupled with PTT could be an approach that helps to increase the bactericidal effect of certain therapeutic molecules. Indeed, we are interested in metal-based nanoparticles of gold and silver. The combination of the two metals (Au and Ag) in a single nanomaterial (Au/Ag NP) could cause a synergistic effect of the properties and increase the antibacterial activity. However, the use of silver in combination with gold could open new possibilities for the conjugation of antibacterial silver nanoparticles with various biomolecules via a covalent bond to gold atoms, which will necessarily increase antimicrobial efficacy and targeting sensitivity.

Ag NPs have shown more potent antimicrobial activities. Indeed, several studies have shown the strong antimicrobial activity of Ag NPs with smaller sizes. These NPs can disrupt the bacterial cell membrane, affecting cell penetration, but they cause high toxicity, which impairs their biocompatibility.

Au NPs also have a role in antimicrobial activity. The determinants of the antimicrobial effects of Au NPs are widely studied, such as shape, size, concentration, and coating agent. Especially due to the electrical and optical properties of Au NPs, they have been given more consideration. One of the predominant properties is their localized LSPR. This occurs when electrons on the surface of Au NPs interact with electromagnetic radiation, thereby producing an LSPR. This characteristic gives them an important role in various applications such as biosensors in the case of different phototherapies.

In addition, we have noted that these nanocompounds have been grafted to molecules (PDA, ICG, or carbon-based materials), which serve to improve their plasmonic and photothermal characteristics, or bound to bacterial peptides for targeted treatment.

Some studies have also revealed the antibacterial effects of bimetallic nanostructures and found that the nanoparticles of gold and silver coupled with other metals (Cu or Zn) improve their antibacterial activity. It would therefore be interesting to develop a theranostic strategy to target superficial and endogenous infections by combining chemotherapy, PTT, and PTAT to respond to antibiotic resistance issues. Physically loaded chemotherapy drugs can leak or be released unexpectedly, resulting in short-term treatment for patients. Finding a carrier capable of chemically grafting chemotherapeutic drugs through dynamic reversible chemical bonds and chemically grafting and physically loading PTAT synchronously is needed to design this innovative strategy.

## Figures and Tables

**Figure 1 microorganisms-11-02084-f001:**
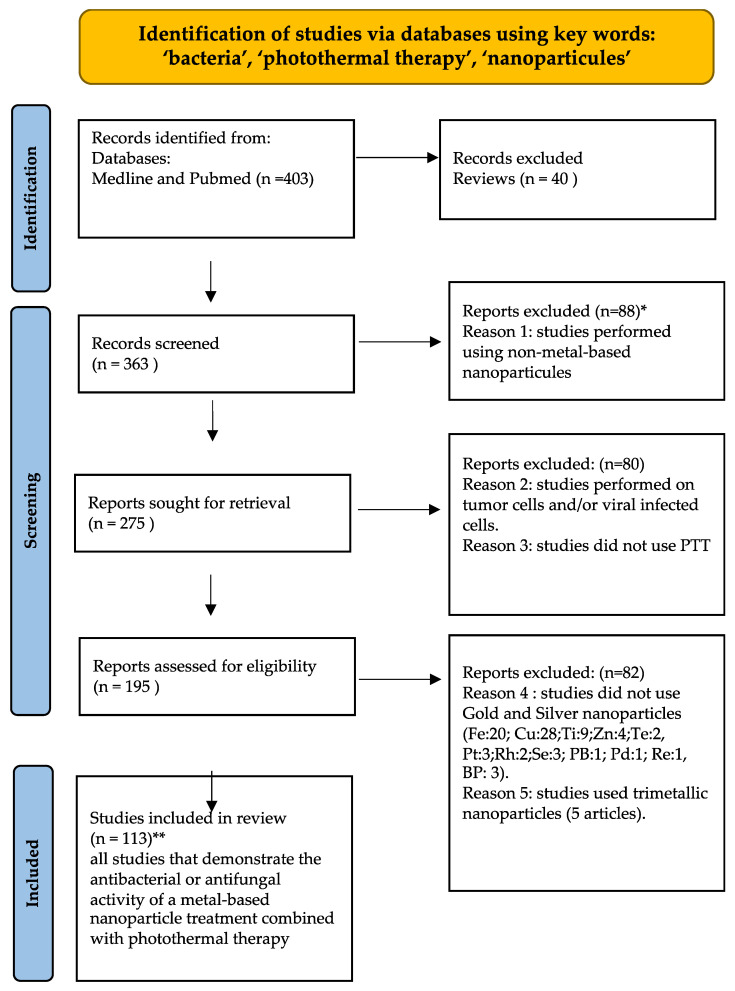
PRISMA flow diagram of photothermal therapy combined with metal-based nanomaterials for the treatment of microbial infections. * two duplicate excluded articles. ** five duplicate included articles.

**Figure 2 microorganisms-11-02084-f002:**
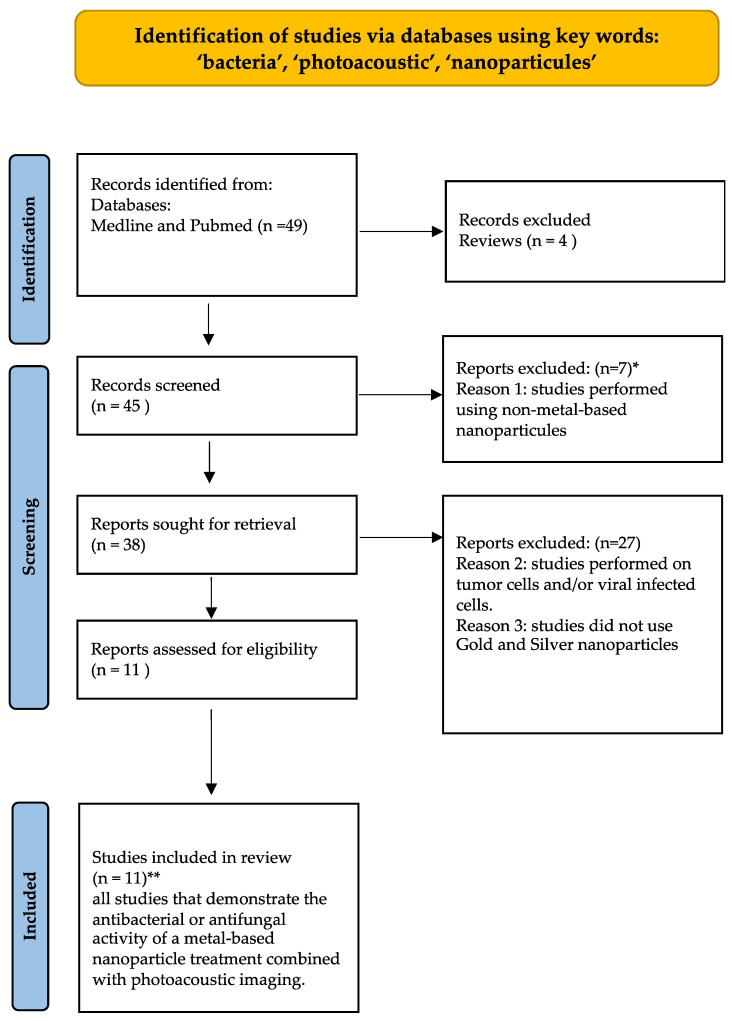
PRISMA flow diagram of photothermal and photoacoustic therapy combined with metal-based nanomaterials for the treatment of microbial infections. * two duplicate excluded articles. ** five duplicate included articles.

**Table 1 microorganisms-11-02084-t001:** Au- or Ag-based nanostructures combined with PTT. The table presents the results of a literature review on monometallic nanoparticles based on Au or Ag and used as antimicrobial agents in a treatment combined with PTT and/or PTD.

Nanostructure	PT	Light Source	PT Parameters	Bacteria or Type of Infection	References
CG/PDA@Ag	PTT	Laser	808 nm (1 W/cm^2^) 3 min (37 to 49.1 °C)	* E. coli* * S. aureus*	[23]
MX@AgP nanoparticle (NPs)	PTT	Laser	808 nm (1.5 W/cm^2^) 10 min	*S. aureus* (ATCC 25923) *E. coli* (ATCC 25922)	[32]
AuNPTs (Gold nanoplates)	PTT	Laser	808 nm (1 W/cm^2^) 3 min	MRSA	[33]
GQD–AgNP (Ag nanoparticle-conjugated graphene quantum dots)	PTD, PTT	Laser	450 nm (14.2 mW/cm^2^) 10 min (40 °C)	*E. coli* *S. aureus*	[35]
FGO–Ag (Graphene oxide–silver)	PTT	Laser	808 nm (2.0 W/cm^2^) 5 min	*E. coli* *S. aureus*	[36]
Ag@BP nanohybrids	PTT	Laser	808 nm (0.8 W/cm^2^) 5 min	MRSA	[38]
Tri–Ag Silver triangular nanoparticles	PTT	Laser	808 nm (1.3 W/cm^2^) 10 min	*E. coli* *E. coli* (ESBL) *S. aureus* MRSA	[39]
Ag NPs-incorporated quaternized chitin (DQCA) nanomicelle	PTT	Laser	660 nm (1.0 W/cm^2^) 10 min	*S. aureus* *E. coli*	[40]
AgNPs/POM-PDA (a three-in-one bactericidal flower-like nanocomposite–Ag nanoparticles/phosphotungstic acid–polydopamine nano-flowers)	PTT	Laser	808 nm (0.75 W/cm^2^) 6 min	*E. coli* *S. aureus*	[41]
Polydopamine (PDA) coating-reduced Ag nanoparticles (AgNPs)	PTT	Laser	808 nm 10 min	*E. coli* * S. aureus*	[42]
Ag-doped Mo_2_C-derived polyoxometalate (AgPOM) nanoparticles urea, gelatin, and tea polyphenols (TPs)	PTD, PTT	Laser	1060 nm (1.0 W/cm^2^) 10 min	* S. aureus*	[43]
SABA/Borax/PDA@AgNPs hydrogel	PTT	Laser	808 nm, 5 min (≤45 °C)	*S. aureus* *E. coli*	[44]
AgNPs@TA (Hyaluronic acid–tyramine (HT) hydrogel loaded with antioxidant and photothermal silver nanoparticles (AgNPs))	PTT	Laser	808 nm (0.92 W/cm^2^) 10 min	*S. aureus* *E. coli*	[45]
Soie-GOx-Ag @ G, SGA (Silk-GOx-Ag@G, SGA (glucose oxidase (GOx) is embedded in Ag graphitic nanocapsule (Ag@G))	PTT	Laser	808 nm (5 W/cm^2^) ( 60 °C)	* S. aureus* MRSA	[46]
Ag_2_O_2_ NPs (Silver peroxide nanoparticles)	PTT	Laser	808 nm (0.7 W/cm^2^) 10 min	MRSA *S. aureus* *E. coli* *P*. *aeruginosa*	[47]
AgBiS_2_ QDs (Silver bismuth sulfide quantum dots)	PTD, PTT	Laser	808 nm (1.6 W/cm^2^) 10 min	* E. coli* * S. aureus*	[48]
AgNC/GSH-rGO (Ag nanoclusters, graphene oxide (rGO), glutathione (GSH))	PTT	Laser	808 nm (2 W/cm^2^) 5 min	*E. coli* *S. aureus*	[49]
GO–HA–AgNPs (a hyaluronidase, silver nanoparticles (AgNPs) and graphene oxide (GO))	PTT	Laser	808 nm (1.0 W/cm^2^) 2 min	*S. aureus*	[50]
Carbon fiber oxide (FCO)/Ag composite	PTT	Laser	808 nm (2 W/cm^2^) 5 min	*E. coli* *S. aureus*	[51]
Mesoporous silica nanospheres (HMSN)/silver nanoparticles (Ag NPs)/vancomycin (Van)/hemin (HAVH)	PTT	Laser	808 nm (1 W/cm^2^) 10 min	*MRSA*	[52]
AuNSs (Gold nanoclusters)	PTT	Laser	808 nm (1 W/cm^2^) 300 s (53.1 °C)	*S. aureus*	[53]
PHMB@Au NPs (Polymer polyhexamethylene biguanide (PHMB, with bactericidal and anti-biofilm functions) hybrid gold nanoparticle (Au NPs))	PTT	Laser	808 nm (2.0 W/cm^2^) 10 min ( 45.0 °C) (4.5 µg/mL), 58.4 °C (9.0 µg/mL), 65.2 °C (18.0 µg/mL)	* S. aureus*	[54]
IgG-AuNPs (Gold nanoparticles)	PTT	Laser	808 nm (2 W/cm^2^) 10 min	MRSA	[55]
AuNPs Gold nanoparticles	PTT	Laser pulses	420–570 nm, 12 ns, 0.1–5 J/cm^2^, 100 pulses	*S. aureus*	[56]
PDA-AuNPs (Polydopamine–gold nanoparticles)	PTT	Laser	808 nm (1 W/cm^2^) 15 min (55 °C)	*S. aureus* (ATCC 6538) MRSA *E. coli* (ATCC 25922)	[57]
GNS/HPDA JNPs (Gold nanostar/hollow polydopamine Janus nanostructure)	PTT	Laser	808 nm (1.5 W/cm^2^) 5 min (125 µg/mL)	*E. coli* (ATCC 25922) *S. aureus* (ATCC 29213) MRSA (ATCC 43300)	[58]
PDA@Au-HAp NPs (a polydopamine (PDA) coating on hydroxyapatite (HAp) incorporated with gold nanoparticles (Au-Hap))	PTT	Laser	808 nm (1.0 W/cm^2^) 10 min	*E. coli* *S. aureus*	[59]
Apt@AuNPs (DNA aptamer-functionalized gold nanoparticles)	PTT	Laser	808 nm (1.1 W/cm^2^) 2 min	MRSA (ATCC 43300)	[60]
AuNPs-ICG (Porous gold nanoparticles–indocyanine green)	PTD, PTT	Laser	808 nm	* S. aureus*	[61]
AuNPs_ICG	PTT	Laser	808 nm (1 W/cm^2^) 1 min	*E. coli* (ATCC 8393) *S. aureus* (ATCC 6538P)	[62]
AuNRs (Gold nanorods)	PTT	Laser	810 nm (6.3 W/cm^2^) 10 min	*E. coli* *E. coli*/AuNRs	[63]
AuNR@P(NIPAM-AA-MAA) (N-isopropyl acrylamide (NIPAM), acrylic acid (AA), and N-allylmethylamine (MAA)–gold nanorods (AuNRs))	PTT	Laser	808 nm (1.0 W/cm^2^) 10 min	*E. coli* *S. aureus*	[64]
AuNC@NO	PTT	Laser	808 nm (0.5 W/cm^2^) 5 min	MRSA	[65]
PDG@Au-NO/PBAM (dopamine-co-glucosamine,Gold, Nitric oxide, phenylboronic acid, and acryloylmorpholine)	PTT	Laser	808 nm (1.0 W/cm^2^) 10 min	MRSA (ATCC BAA-40) TREC (ATCC ER2738)	[66]
PDA-AuNCs (Antibody-conjugated, polydopamine (PDA)-coated gold nanocages (AuNCs))	PTT	Laser	808 nm (0.8 W/cm^2^) 10 min	MRSA *P*. *aeruginosa* (ATCC 27317)	[67]
SNP@MOF@Au-Mal nanogenerator	PTT	Laser	808 nm (1.5 W/cm^2^) 10 min	* P. aeruginosa*	[68]
Gold asymmetrically functionalized mesoporous silica half-shell nanoswimmer (HSMV)	PTT	Laser	650 nm (1.5 W/cm^2^) 10 min	*S. aureus*	[69]
Au@Van NPs (Vancomycin-immobilized gold nanoparticles)	PTT	Laser	808 nm 5 min	* Vancomycin-resistant Enterococci * (VRE)	[70]
(gold (Au^1^)–UCNP–gold (Au^2^)) (Gold sandwich UCNP nanocomposites)	PTT	Laser	980 nm (0.2 kW/cm^2^) 20 min	*E. coli* (BCRC 12438) *S. aureus* (BCRC 10 780)	[71]
MPBA/pAu chip (4-mercaptophenylboronic acid (4-MPBA)/Au)	PTT	Laser	808 nm 10 min	*S. aureus* (ATCC 29213) *E. coli* (ATCC 8739)	[72]
Da-Au_n_NFs (Daptomycin–gold nanoflowers)	PTT	Laser	808 nm (1.75 W/cm^2^) 10 min	*E. coli* *S. aureus*	[73]
AuNSs@Van (Vancomycin (Van)-modified gold nanostars (AuNSs))	PTT	Laser	808 nm (2.5 W/cm^2^) 10 min	*S. aureus* (1213P46B) *S. aureus* (AB91093) MRSA (011P6B5A) ampicillin-resistant *E. coli* (PCN033) *E. coli* (AB 93154)	[74]
Van-TCO-NHS-AuNPs (Vancomycine-E-cyclooct-4-enyl-2,5-dioxo-1-pyrrolidinyl carbonate–gold nanoparticle)	PTT	Laser	808 nm (2 W/cm^2^) 5 min	*Bacillus subtilis* (ATCC 6633) *S. aureus* (ATCC 700698) *Enterococcus faecalis* (ATCC 29212) *E. coli* (ATCC 53868)	[75]
DNase-AuNCs (Eoxyribonuclease (DNase)-functionalized gold nanoclusters (AuNCs))	PTT	Laser	808 nm (2 W/cm^2^) 10 min	*E. coli* *S. aureus*	[76]
TC-AuNSs (Thiol chitosan-wrapped gold nanoshells)	PTT	Laser	808 nm (0.95 W/cm^2^) 5 min	*S. aureus* *E. coli* *P*. *aeruginosa*	[77]
AuNPs/CS-Cur	PTD, PTT	Laser	405 + 808 nm 5 min	*E. coli* *P*. *aeruginosa* *Bacillus subtilis* *S. aureus*	[78]
TBO-AuNPs (Toluidine blue O (TBO) and gold nanoparticles (AuNPs))	PDT, PTT	helium-neon laser light light-emitting diode (LED)	PDT: 633 nm, 530 nm (85 mW)	* E. coli* * Bacillus cereus *	[79]
Gold nanoparticle (AuNP)-targeted pulsed laser therapy + ATB	PTT	LED	530 nm (85 mW) 5 min	MRSA (SA5120) MDR *P*. *aeruginosa* (PA 60–65)	[80]
Au/i-form/Au/s-form (a gold-binding peptide motif displayed on the pVIII major coat protein templated Au nanoparticles)	PTT	Laser	532 nm (0, 100, 200, and 300 mW/cm^2^) 20 min	*E. coli* (K12 ER2738)	[81]
Gold-nanoparticle-decorated porous silicon nanopillars	PTT	Laser	808 nm (1.25 W/cm^2^) 10 min	*E. coli* (ATCC 25922) *S. aureus* (ATCC 29213)	[82]
surface-adaptive gold nanoparticles (AuNPs) zwitterionic self-assembled monolayers 11-mercaptoundecanoic acid (HS-C10-COOH) and (10-mercaptodecyl) trimethylammonium bromide (HS-C10-N4)	PTT	Laser	808 nm (0.91 W/cm^2^) 10 min	MRSA (ATCC 43300)	[83]
AuNS@PEG-SH (gold nanoparticles (AuNP) coated with polyethylene glycol)	PTT	CW laser	532 nm (60 mW) 5 min	* E. coli*	[84]
Gold nanorods (AuNRs) with (200) plane and gold nanobipyramids (AuNBPs)	PTT	Laser	808 nm (1.0 W/cm^2^) 10 min	* E. coli*	[85]
Phanorods (phages to gold nanorods)	PTT	Laser	808 nm (3.0 W/cm^2^) 10 min	* E. coli* * P. aeruginosa* * Vibrio cholerae *	[86]
AuNR@C-At5 gold nanorods (AuNRs)/peptide	PTT	Laser	808 nm (2.5 W/cm^2^) 10 min	*E. coli* (ATCC 25922) *S. aureus* (ATCC 25923)	[87]
Au@CDs composite nanoparticles comprised of gold nanoparticles (AuNPs) and carbon dots (N,S-CDs)	PTT	Laser	808 nm (3.0 W/cm^2^) 10 min (50 °C)	*S. aureus* (ATCC 25923) *E. coli* (ATCC 25922)	[88]
SWCNT-AuNPs (Monoclonal antibody-conjugated sphere-shaped gold nanoparticles were combined with single-walled carbon	PTT	LED	670 nm (2 W/cm^2^) 15 min	MDR *Salmonella typhimurium* DT104	[89]

**Table 2 microorganisms-11-02084-t002:** Bimetallic (Au or Ag)-based nanostructures combined with PTT. The table presents the results of a literature review on bimetallic nanoparticles based on Au or Ag and used as antimicrobial agents in treatment combined with PTT.

Bimetallic Nanostructure	PT	Light Source	PT Parameters	Bacteria or Type of Infection	References
HSKAu(rod) (Hybrid bactericidal material, gold nanorod-covered kanamycin-loaded hollow SiO_2_)	PTT	Emitting diode laser	785 nm (120 mW) 20 min	* E. coli* BL21	[92]
AuAg-PC procyanidins	PTT	Laser	808 nm (2.5 W cm^2^) 10 min	*Porphyromonas gingivalis* (ATCC33277)	[93]
Au-AgNPs	PTT	Laser	808 nm (1 W/cm^2^) 5 min	*E. coli* (K-12 strain, WT) *S. epidermidis* (ATCC 12228) *P. aeruginosa* (ATCC 27853)	[94]
Au-AgNPs	PTT	Laser	808 nm (2 W/cm^2^) 5 min	*S. aureus*	[95]
Au/AgNCs (Gold–silver hybrid nanocage)	PTT	Laser	808 nm (1 W/cm^2^) 10 min	Multidrug-Resistant *Acinetobacter baumannii* (MDR-AB)	[96]
Sa-M-AuAgNC (Gold–silver nanocage (AuAgNC))	PTT	Laser	808 nm (1.0 W/cm^2^) 5 min	*E. coli* (ATCC 43888) *S. aureus* (ATCC BAA-1721)	[97]
NiO NPs@AuNPs@Van (NAV)	PTT	Laser	808 nm (1.8 W/cm^2^) 10 min	MRSA	[98]
HydrogelGFA/PDA@Ag NPs: PDA@Ag NPs_ADA_gel Cat	PTT	Laser	808 nm (1.3 W/cm^2^) 10 min	*S. aureus* *E. coli*	[99]
Au NCs@PCN (Gold nanoclusters modified with zirconium-based porphyrin metal–organic frameworks)	PTT	Laser	808 nm (1 W/cm^2^) 10 min (56.2 °C)	*S. aureus* (ATCC 25923) MRSA (ATCC 43300) *E. coli* (ATCC 25922) *Ampr E*. *coli* (ATCC 35218)	[100]
Ag@Au-Ce6 NPs (silver–gold alloy nanoparticles immobilized with the photosensitizer molecule Ce6)	PTT	Laser	808 nm (800 mW/cm^2^ for 5 min) and a 660 nm laser (200 mW/cm^2^ for 5 min)	*S. aureus* (ATCC 25923) *E. coli* (ATCC 25922)	[101]
α-Fe_2_O_3_@Au/PDA core/shell nanoparticles	PTT	Laser	808 nm (2 W/cm^2^) 5 min	* E. coli* * S. aureus*	[102]
BGN-Fe-Ag_2_S (Ag_2_S nanodots conjugated Fe-doped bioactive glass nanoparticles)	PTT	Laser	808 nm laser (1 W/cm^2^) 10 min	* S. aureus* (ATCC 43300)	[103]
Au-ZnO-BP nanocomposite (phosphorus (BP)-based non-damaging near-infrared light-responsive platform conjugated with ZnO and Au nanoparticles)	PTT	Laser	808 nm (2.5 W/cm^2^) 5 min	*S. aureus* (ATCC 25923) MRSA clinical isolates	[104]
Ag-Bi@SiO_2_ NPs (mesoporous silica supported silver–bismuth nanoparticles)	PTT	Laser	808 nm (1 W/cm^2^) 15 min	MRSA	[105]
DTTC AuAgNSs (3,3′-diethylthiatricarbocyanine iodide (DTTC)-conjugated gold–silver nanoshells)	PTT	Laser	808-nm (1.0 W/cm^2^) 10 min	*E. coli* (ATCC 25922) *E. coli* (ESBL) *S. aureus* (ATCC 6538) MRSA	[106]
Ag/ZnO/rGO (Silver, Graphene oxide, zinc oxide)	PTT	*Xenon lamps*	rapid microwave irradiation	*S. aureus* (SA113) *E. coli* (K12)	[107]
AuPtNDs (Gold–platinum nanodots)	PTT	Laser	808 nm (1 W/cm^2^) 15 min	* E. coli* * S. aureus*	[108]
AuNRs/Ti (Gold nanorods–Titanium)	PTT	Laser	808 nm (0.5 W/cm^2^) 20 min	*E. coli* (ATCC 25922) *P*. *aeruginosa* (ATCC 27853) *S. aureus* (ATCC 25923) *S*. *epidermidis* (ATCC 12228)	[109]
MOF/Ag-derived nanocomposite (MOF-derived nanocarbon consisting of metallic zinc and a graphitic-like carbon framework is first synthesized, and then Ag nanoparticles (AgNPs))	PTT	Laser	808 nm (3 W/cm^2^) 10 min	*E*. *coli* *S. aureus*	[110]
ICG@ZIF-8/PDA/Ag	PTT	Laser	808 nm (1.5 W/cm^2^) 20 min	*E. coli* *S. aureus*	[111]
Ag_2_S@ZIF-Van NS	PTT		808 nm (1 W/cm^2^)	*S. aureus*	[112]
Ag+-GCS-PDA@AuNRs (Silver Polydopamine (PDA)-coated gold nanorods (AuNRs))	PTT	Laser	808 nm (0.5 W/cm^2^) 7 min	MRSA *E. coli*	[113]
AuAgCu_2_O-BS NPs (AuAgCu_2_O-bromfenac sodium nanoparticles)	PTT	Laser	808 nm (0.75 W/cm^2^) 10 min	MRSA	[114]
AuAgCu_2_O NS	PTT	Laser	808 nm (2.55 W/cm^2^) 5 min	*E. coli* (ESBL ATCC 35218) MRSA (ATCC 43300)	[115]

**Table 3 microorganisms-11-02084-t003:** Au- or Ag-based nanostructures combined with PTAT (and PTT). The table presents the results of a literature review of studies involving a metallic nanocluster based on Au or Ag for theranostic purposes, carried out using the techniques of PTT and PTAT.

Nanostructures	PT/PTAT	Light Source	PT/PTAT Parameters	Bacteria or Type of Infection	References
AgCuE NPs (Ethylenediaminetetraacetic acid (EDTA)-modified AgCu_2_O nanoparticles)	PDT/PTT/PTAT	Laser	808 nm (0.25 W/cm^2^) 5 min	* C. albicans*	[24]
GP-dAuNPs@Ce6 (Glucose polymer (GP)-modified gold nanoparticles through ATP-binding cassette (ABC))	PTT/PTAT	Lasers	405 nm (1 W/cm^2^) 25 min 660 nm (12 mW/cm^2^) 5 min 808nm (1 W/cm^2^) 5 min	*E. coli* (ATCC 11303) *S. aureus* *Micrococcus luteus* (BNCC 102589) *P*. *aeruginosa* (BNCC 125486)	[116]
Silver nanoparticles	-		-	* - *	[117]
BP Ve-Ag+ QD (Silver-ion-coupled black phosphorus (BP) vesicle quantum dot (QD))	PTT/PTAT	Laser	660 nm (150 mW/cm^2^)	*E. coli* (BNCC 133264) *S. aureus* (ATCC 6538)	[118]
AuNPs@P1 (Peptide modified gold nanoparticles)	PTAT	Laser	710 nm	*S. aureus* (ATCC 6538) *E. coli* (ATCC 25922)	[119]
AuNS@Ab (Gold nanostars@H. pylori-antibodies nanoprobes)	PTT/PTAT	Laser	PTAT: 790 nm (3 W/cm^2^) PTT: 790 nm (1 W/cm^2^) 8 min	* Helicobacter pylori *	[120]
Au/AgNRs (Au/Ag nanoparticles by coating AuNRs with silver (Ag))	PTAT	Pulsed laser	800 nm at 30 MHz	MRSA * E. coli*	[121]
AuNPs (Gold nanoparticle)	PTAT	Pulsed laser	532 nm, an 8 ns pulse duration pulse repetition rate of 1 Hz	MRSA (ATCC 33591) MSSA (ATCC 29213)	[122]
AuNRs (gold nanorods)	PTAT	Laser-induced photoacoustic waves	710 nm and 1 J/cm^2^	* S. aureus* (ATCC 49230)	[123]
Gold nanorods (AuNRs) golden carbon nanotubes (AuNTs) silica-coated magnetic MNPs (siMNPs)	PTT/PTAT	Laser-induced photoacoustic waves Pulsed laser	PTAT: (50 mJ/cm^2^) at 671 nm PTT: 1-h laser exposure, laser fluence of 0.8 mJ/cm^2^ at 850 nm and a pulse rate of 10 Hz	*S. aureus*	[124]
Gold nanoparticles (AuNRs) nanoshells (AuNSs)	PTAT	Laser	830 nm, 100 mJ/cm^2^	*S. aureus* (ATCC 49230) *E. coli* K12	[125]
AuNR-SiO_2_-Cu_7_S_4_	PTT/PTAT	Laser	1064 nm (0.75 W/cm^2^), 5 min	MRSA (ATCC43300)	[126]
PTNP gold in a gold cage photothermal nanoparticles	PTT/PTAT	Laser	808 nm (2 W/cm^2^), 10 min	*Streptococcus mutans* UA159 (ATCC 700610) MRSA (ATCC43300)	[127]

## Data Availability

Not applicable.

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
