# Peer review of "Photothermal/Photoacoustic Therapy Combined with Metal-Based Nanomaterials for the Treatment of Microbial Infections"

_microorganisms, 2023, doi:10.3390/microorganisms11082084_

Round 1

Reviewer 1 Report

The manuscript "Photothermal/photoacoustic therapy combined with metal-based nanomaterials for treatment of microbial infections" offers a comprehensive description of using PTT and PDT to fight against antibiotic resistant bacterial infections.

The authors provide a clear and useful review. It is clearly of use for researchers in this particular niche.

Author Response

Reviewer 1

The manuscript "Photothermal/photoacoustic therapy combined with metal-based nanomaterials for treatment of microbial infections" offers a comprehensive description of using PTT and PDT to fight against antibiotic resistant bacterial infections.

The authors provide a clear and useful review. It is clearly of use for researchers in this particular niche.

Authors’ response

We are grateful to reviewer 1 for their comments.

Reviewer 2 Report

microorganisms-2527194

Review of the article titled: 

“Photothermal/photoacoustic therapy combined with metal-based nanomaterials for treatment of microbial infections.”

by Nour Mammari, Raphaël Emmanuel Duval

In Microorganisms (ISSN 2076-2607)

Round 1

The abstract perfectly underlines the relevance of the presented research. The paper has been slightly improved, enlarged, and several statements should be clarified. This review should be reevaluated after minor revision.

1.     The literature source, if possible, should include information concerning patents.

2.     The recent articles for the last four months information should be checked and included.

3.     The paragraph about the stability and purity of metal-based nanomaterials for photothermal or photoacoustic therapy should be evaluated.

4.     Also, several statements on the reproducibility of the synthesis can be added.

5.     I highly recommend including the figures from the articles.

6.     I think the photoacoustic contrast agents can be highlighted in a separate paragraph.

7.     Introduction. In the introduction section, it is necessary to clearly define the range of issues considered in the review and the time period. Here it is also required to note whether there are reviews on similar topics in the world literature of the last ten years. List them and indicate the principal differences between this review and the ones already available. 

-       The purpose of the Introduction is to attract the reader of Microorganisms (MDPI) by what he can find here in contrast to other similar publications.

8.     Several recent reviews and research articles used an NPs approach. Please use the appropriate references to improve the quality of the work.

-       http://dx.doi.org/10.3367/UFNe.2021.05.038976;

-       https://doi.org/10.1016/j.mattod.2022.11.011;

-       https://doi.org/10.1016/j.pacs.2021.100281 etc.

9.     The review's conclusion should be detailed about the state of the field of science to date, with a brief description of achievements, shortcomings, and prospects for future development.

10.  The authors should be double-checked for English grammar and spelling.

Author Response

Reviewer 2

The abstract perfectly underlines the relevance of the presented research. The paper has been slightly improved, enlarged, and several statements should be clarified. This review should be reevaluated after minor revision.

Authors’ response

We would like to thank the Reviewer for their comments. We considered each comment and made changes where possible, based on the purpose of this literature review and the data reported in the cited studies.

  1. The literature source, if possible, should include information concerning patents.

Authors’ response

These patent ‘s references have been added in ‘introduction’ part:

·       Metallic nanoparticles, preparation and uses thereof: EP3178494A1

·       Nanoparticle composition and methods of making the same: US9378861B2

  1. The recent articles for the last four months information should be checked and included

Authors’ response

The literature review has been updated to July 28, 2023.

Prisma Flow diagrams (Figures 1 and 2), Tables 1 and 2 have been corrected according the references added.

These paragraphs were added:

July 2023

With the aim of having a system for delivering an antibacterial metal treatment. Hollow mesoporous silica nanospheres (HMSN) were designed and loaded with silver nanoparticles (Ag NPs), vancomycin (Van) and hemin (HAVH) for the elimination of bacteria and the abscesses treatment. However, this nanocomplex is photosensitive. Following NIR light irradiation (808 nm, 1.0 W/cm) (45°C) for 10 min at a concentration of 250 µg m/L, hemin is released from the mesopores of HMSN thus triggering the opening of the pores and the release of preloaded Ag+ and Van. A synergistic photothermo-chemotherapy activity of the nanocomplex was thus obtained to fight against MRSA, both in vitro and in vivo. The results showed also that hemin, which possessed intrinsic ROS scavenging activity, could attenuate the inflammatory response at the treatment site and benefit the wound healing process [52].

Recently, hollow silver-gold alloy nanoparticles immobilized with the photosensitising molecule Ce6 (Ag@Au-Ce6 NPs) integrated with PTT at 808 nm (800 mW/cm2 for 5 min) and a 660 nm laser (200 mW/cm2 for 5 min) are designed to accelerate wound healing in the context of infection by S. aureus (ATCC 25923) or E. coli (ATCC 25922). As a result of the heat effect provided by NIR laser, Ag@Au-Ce6 NPs at 0.25 nM could effectively kill the free and colonized bacteria on the surface of the injured skin via the release of ROS, thereby promoting the vascularization of the skin epithelium and wound healing [102].

March 2023

Another model of NO nanogenerator has been developed for biofilm eradication. PDG@Au-NO/PBAM is composed of the heat-sensitive NO-donating conjugated AuNPs on cationic poly (dopamine-co-glucosamine) (PDG@Au-NO) nanoparticles, and an anionic copolymer of phenylboronic acid and acryloylmorpholine (PBAM) which are used for a shell. However, according to authors PDG@Au-NO/PBAM seems to be a good generator of NO, when it reaches the biofilm, its surface charge would be positive after the dissociation of the shell and the exposure of the cationic nucleus, which will allow them to s infiltrate and accumulate in the depth of the biofilm. At concentration of 150 μg/mL and under NIR irradiation at 808 nm, 1.0 W/cm2 for 10 min, PDG@Au-NO/PBAM could sustainably generate NO to disrupt biofilm integrity under hyperthermia (54°C). This will serve to effectively eradicate the biofilm of resistant bacteria, MRSA (ATCC BAA-40) and TREC (ATCC ER2738). Moreover, the in vitro cytotoxicity of PDG@Au–NO/PBAM nanogenerator with mouse embryonic fibroblast cells (NIH3T3) showed that PDG@Au–NO/PBAM at a concentration of 150 μg/mL could promote cell survival and proliferation through protection against apoptosis [66] .

May 2023

Recently, dual plasmonic AuNR-SiO2-Cu7S4 antibacterial nanomotors have been developed with a Janus configuration predicted by the proliferation of copper-rich Cu7S4 nanocrystals at a high curvature site of AuNRs with a diameter of ∼95 nm in length and ∼18 nm. Upon exposure to NR light at 1064 nm (0.75 W/cm2) for 5 min at a concentration of 100 μg/mL, a local photothermal field is formed near the AuNR–Cu7S4 interface, thus resulting in improved photothermal performance and antibacterial activities of photocatalytic ROS generation. In vivo treatment performed by AuNR-SiO2-Cu7S4 synchronous autonomic movement triggered by NIR light and synergistic photothermal/photocatalytic/photoacoustic antibacterial nanomotors improves transdermal penetration and effectively treats MRSA (ATCC43300) infections at higher efficacy by 98%. In addition, excellent biosecurity has been assessed. Cellular cytotoxicity tests demonstrated good biocompatibility of the nanomotors [128].

An effective theranostic anti-biofilm agent PTNP based on Au@Au core-cage structures has been developed allowing rapid photoablation and biofilm disruption. In vitro and in vivo treatment with 3.75 mg of PTNP accompanied by NIR irradiation at 808 nm (2 W/cm2) for 10 min against Streptococcus mutans (S. mutans), UA159 (ATCC 700610) and MRSA (ATCC43300) demonstrates efficacy of photothermal conversion, a capacity in imaging of biofilm bacteria and their rapid elimination. This structure is non-toxic and biocompatible [129].

  1. The paragraph about the stability and purity of metal-based nanomaterials for photothermal or photoacoustic therapy should be evaluated.

Authors’ response

The paragraph has been modified:

the light source has been added, 3 studies have been added after the update of the bibliography, the paragraph dedicated to the AgCuE NPs molecule has been modified, the size of the nanoparticles, the concentration and the biocompatibility have been added when these parameters are indicated in the studies.

  1. Also, several statements on the reproducibility of the synthesis can be added.

Authors’ response

This review of the literature did not deal with the methodology for the synthesis of metallic nanocompounds or their characteristics. This requires further consideration and mastery of the chemical synthesis and photophysical characteristics of metal nanocomplexes.

  1. I highly recommend including the figures from the articles.

Authors’ response

Each study contains figures relating to the synthesis of the metallic nanoparticle as well as its antimicrobial application. It would be difficult to take figures from all the articles and include them in the review. This present review contains three tables which summarize all the cited studies with their bibliography references, which will facilitate access to the cited article.

  1. I think the photoacoustic contrast agents can be highlighted in a separate paragraph.

Authors’ response

Metal nanocomplexes and alloys have been reported in the photoacoustic section. When a nanocomplex is grafted onto an exp: ICG fluorophore, this is indicated in the dedicated section.

  1. In the introduction section, it is necessary to clearly define the range of issues considered in the review and the time period. Here it is also required to note whether there are reviews on similar topics in the world literature of the last ten years. List them and indicate the principal differences between this review and the ones already available. 

Authors’ response

Published literature reviews do not necessarily deal with the same subject addressed in this present literature review.

However, our main objective is to have a general view on nanoparticles and nanocomposites designed based on Ag and Au (including AuAg alloys) for antibacterial treatment coupled with PTT. Literature reviews published today generally present the different strategies used to eliminate bacterial biofilm and increase wound healing, to eliminate stomach infection caused by the bacterium Helicobacter pylori, to treat dental caries and in the case of cancer treatment (which is not the subject of this present literature review).

In a secondary objective we are interested in determining which metallic nanocomposites (based on Au and Ag) are used for photoacoustic in the context of a bacterial infection. Indeed, this is the first literature review that addresses this objective in the context of bacterial infection. Following your request, we have included two literature reviews in the ‘introduction’ part.

These paragraphs are added in introduction section:

In addition, the incorporation of metal ions like Ag+ Cu2+, Zn2+, Co2+, Fe3+and Pb 2+ could lead to an increase in the innate cytotoxicity response of these materials and cause a synergistic effect that bacterial infection [13]. Doi : 10.3390/pharmaceutics15051521

It has been described that metal nanoparticles coupled with polymers have an effective approach to remove bacterial biofilm to internalize and disrupt surface functionalization to increase biocompatibility. By coupling this treatment to PDT, the latter could act by killing the bacteria hidden in the biofilm via the destruction of the channels engaged in the transport of nutrients towards the central region and the loss of the integrity of the biofilm. However, these photothermal effects presented some side effects by damaging normal tissues. The combination of PDT and PTT has emerged to maximize efficacy while minimizing side effects [26].Doi : 10.3389/fcimb.2022.1003033

In photoacoustic, plasmonic nanoparticles are widely used. They lead to a strong light/matter interaction. It was determined that the absorption efficiency of metallic nanoparticles should be higher than that of dye molecules. This results in efficient generation of photoacoustic signals. These signals are generated by exciting either the signals of the constituent molecules of living cells, or those of exogenous contrast agents. However, the optimal excitation light wavelength can be selected depending on the size, shape and composition of the nanoparticle in question. Moreover, this therapy could be targeted by the control of the surface chemistry which confers a selective functionalization of the metallic nanoparticle [30] DOI: 10.1016/j.pacs.2021.100281

  1. Several recent reviews and research articles used an NPs approach. Please use the appropriate references to improve the quality of the work.

Authors’ response

- http://dx.doi.org/10.3367/UFNe.2021.05.038976;=> we don’t have access at this scientific article.

- https://doi.org/10.1016/j.mattod.2022.11.011=> This article has been excluded because it is a PhotoTheranostic therapy in the context of cancers.

- https://doi.org/10.1016/j.pacs.2021.100281 etc.=> This review has been added in the 'introduction' part.

In photoacoustic, plasmonic nanoparticles are widely used. They lead to a strong light/matter interaction. It was determined that the absorption efficiency of metallic nanoparticles should be higher than that of dye molecules. This results in efficient generation of photoacoustic signals. These signals are generated by exciting either the signals of the constituent molecules of living cells, or those of exogenous contrast agents. However, the optimal excitation light wavelength can be selected depending on the size, shape and composition of the nanoparticle in question. Moreover, this therapy could be targeted by the control of the surface chemistry which confers a selective functionalization of the metallic nanoparticle [30] .

  1. The review's conclusion should be detailed about the state of the field of science to date, with a brief description of achievements, shortcomings, and prospects for future development.

Authors’ response

Conclusion has been changed:

We have seen in this review that researchers are working hard to develop a strategy to tackle antibiotic resistance. Chemotherapy coupled with PTT could be an approach that helps to increase the bactericidal effect of certain therapeutic molecules. Indeed, we are interested in metal-based nanoparticles of gold and silver. The combination of the two metals (Au and Ag) in a single nanomaterial (Au/Ag NP) could cause a synergistic effect of the properties and increase the antibacterial activity. However, the use of silver in combination with gold could open new possibilities for the conjugation of antibacterial silver nanoparticles to various biomolecules via a covalent bond to gold atoms, this will necessarily increase antimicrobial efficacy and targeting sensitivity.

Ag NPs showed more antimicrobial activities. Indeed, several studies show the strong antimicrobial activity of Ag NPs with smaller sizes. These NPs can disrupt the bacterial cell membrane, affecting cell penetration, but they cause high toxicity which impairs its biocompatibility.

Au NPs also have a role in antimicrobial activity. The determinants of the antimicrobial effects of Au NPs are widely studied, such as shape, size, concentration, and coating agent. Especially, due to the electrical and optical properties of Au NPs, they have more consideration. One of the predominant properties is their localized LSPR. This occurs when electrons on the surface of Au NPs interact with electromagnetic radiation, thereby producing an LSPR. This characteristic gives it an important role in various applications such as biosensors in the case of different phototherapies.

In addition, we have contacted that these nanocompounds have been grafted to molecules (PDA or ICG or carbon-based materials), which serve to improve their plasmonic and photothermal characteristics or bound to bacterial peptides for targeted treatment.

Some studies have also revealed the antibacterial effect of bimetallic nanostructures and it turned out that the nanoparticles of gold and silver coupled with other metals (Cu or Zn) improve their antibacterial activity. It is therefore interesting to develop a theranostic strategy to target superficial and endogenous infections by combining chemotherapy, PTT and PTAT to respond to antibiotic resistance issues. Physically loaded chemotherapy drugs can leak or be released unexpectedly, resulting in short-term treatment for patients. Finding a carrier capable of chemically grafting chemotherapeutic drugs through dynamic reversible chemical bonds and chemically grafting and physically loading PTAT synchronously is needed to design this innovative strategy.

  1. The authors should be double-checked for English grammar and spelling.

Authors’ response

English grammar has been checked.

Reviewer 3 Report

1-      Please include the light source used for PT whether light bulbs or laser light in Tables 1, 2, and 3. If a laser light source was used, please specify whether it is CW or pulsed laser light.

2-      Please address in the text the level of safety and biocompatibility of metallic nanoparticles used for bacterial and fungal treatment.

3-      In the text, please investigate the impact of metallic nanomaterial size, shape, and concentration.

4-      Most of the laser systems mentioned in this review are CW lasers, so how about the pulsed laser system (nanosecond, picosecond, or femtosecond lasers light)?

5-      Please discuss in the text the bactericidal effect of metal alloy nanoparticles.

6-      In addition to the References list, other current research on the use of laser light as a PT tool is proposed to be reviewed and included if they are beneficial:

Ø    “Using Femtosecond Laser Light-Activated Materials: The Biomimetic Dentin Remineralization Was Monitored by Laser-Induced Breakdown Spectroscopy.” Medicina, vol. 59,3 591. 16 Mar. 2023,

                   doi:10.3390/medicina 59030591

Ø    “Femtosecond laser attenuates oxidative stress, inflammation, and liver fibrosis in rats: Possible role of PPARγ and Nrf2/HO-1 signaling.” Life sciences vol. 307 (2022): 120877.

                doi:10.1016/j.lfs.2022.120877 

Ø    “The bactericidal efficacy of femtosecond laser-based therapy on the most common infectious bacterial pathogens in chronic wounds: an in vitro study.” Lasers in medical science vol. 36,3 (2021): 641-647.    

                  doi:10.1007/s10103-020-03104-0

Overall, the manuscript appears to be well-written.

Author Response

Reviewer 3

We would like to thank the Reviewer for their comments. We considered each comment and made changes where possible, based on the purpose of this literature review and the data reported in the cited studies.

  1. Please include the light source used for PT whether light bulbs or laser light in Tables 1, 2, and 3. If a laser light source was used, please specify whether it is CW or pulsed laser light.

Authors’ response

The light sources used for each study are added in the Table 1,2 and 3. The CW laser is mainly used.

  1. Please address in the text the level of safety and biocompatibility of metallic nanoparticles used for bacterial and fungal treatment

Authors’ response

The biocompatibility and cytotoxicity have been added in the text when they are indicated in the study.

  1. In the text, please investigate the impact of metallic nanomaterial size, shape, and concentration.

Authors’ response

The size, shape and concentration are added in each dedicated section when these characteristics are indicated in the scientific publication.

The objective of this literature review is to provide an overview of metal nanocomplexes (based on Ag and Au) used to irradiate a bacterial infection with photothermal therapy and photoacoustic imaging. The shape and size of nanoparticles and nanocomplexes is just indicated in the text, but it is not discussed because several parameters can have an impact on the antibacterial activity of these metallic agents. To date, there is no specific size and shape at which the antibacterial activity appears to be effective. The constitution of the nanocomposite, the metal used, the PH, the absorption, the mode of absorption and elimination, the targeting, the biocompatibility... are important parameters to take into consideration.

Previous literature studies have suggested that the infiltration of the antibacterial agent (the nanoparticles) into the skin appears to be enhanced in the size range of approximately 80-200 nm compared to drugs in the liquid state. In addition to the nanoscale effect, since energy-dependent pathways play a key role in nanoparticle uptake into cells, the thermal property and unique motion behavior of nanomotors with NIR- They will improve considerably transdermal penetration.

  1. Most of the laser systems mentioned in this review are CW lasers, so how about the pulsed laser system (nanosecond, picosecond, or femtosecond lasers light)?

Authors’ response

The Light sources are indicated and added in Tables 1,2 and 3. Only studies: 56,123,124,126 use a nanosecond pulse laser.

  1. Please discuss in the text the bactericidal effect of metal alloy nanoparticles.

Authors’ response

To discuss the antibacterial effect of Au/Ag alloys. the following text is added to the "conclusion" part:

“The combination of the two metals (Au and Ag) in a single nanomaterial (Au/Ag NP) could cause a synergistic effect of the properties and increase the antibacterial activity. However, the use of silver in combination with gold could open new possibilities for the conjugation of antibacterial silver nanoparticles to various biomolecules via a covalent bond to gold atoms, this will necessarily increase the effectiveness. antimicrobial and targeting sensitivity.

Ag NPs showed more antimicrobial activities. Indeed, several studies show the strong antimicrobial activity of Ag NPs with smaller sizes. These NPs can disrupt the bacterial cell membrane, affecting cell penetration, but they cause high toxicity which impairs its biocompatibility.

AuNPs also have a role in antimicrobial activity. The determinants of the antimicrobial effects of Au NPs are widely studied, such as shape, size, concentration, and coating agent. Especially, due to the electrical and optical properties of Au NPs, they have more consideration. One of the predominant properties is their localized LSPR. This occurs when electrons on the surface of AuNPs interact with electromagnetic radiation, thereby producing an LSPR. This characteristic gives it an important role in various applications such as biosensors in the case of different phototherapies.”

  1. In addition to the References list, other current research on the use of laser light as a PT tool is proposed to be reviewed and included if they are beneficial:

Authors’ response

Ø    “Using Femtosecond Laser Light-Activated Materials: The Biomimetic Dentin Remineralization Was Monitored by Laser-Induced Breakdown Spectroscopy.” Medicina, vol. 59,3 591. 16 Mar. 2023,

                   doi:10.3390/medicina 59030591

Ø    “Femtosecond laser attenuates oxidative stress, inflammation, and liver fibrosis in rats: Possible role of PPARγ and Nrf2/HO-1 signaling.” Life sciences vol. 307 (2022): 120877.

                doi:10.1016/j.lfs.2022.120877 

These two studies cited below do not encompass the objective of the literature review.

Ø    “The bactericidal efficacy of femtosecond laser-based therapy on the most common infectious bacterial pathogens in chronic wounds: an in vitro study.” Lasers in medical science vol. 36,3 (2021): 641-647.    

                  doi:10.1007/s10103-020-03104-0

This present study is very interesting, but it is about the evaluation of the antibacterial efficacy of Au quantum dots in laser-based PDT and our study essentially deals with studies that used PTT.

Round 2

Reviewer 3 Report

The authors have made reasonable changes to the manuscript in response to my previous suggestions and concerns. In my opinion, the manuscript now has all information and is ready for publication as a regular article in the Journal " microorganisms ".

Overall the manuscript appears to be clearly and carefully written.